# A critical role of VMP1 in lipoprotein secretion

Hideaki Morishita[1‡], Yan G Zhao[2,3†], Norito Tamura[1†§], Taki Nishimura[1#], Yuki Kanda[1], Yuriko Sakamaki[4], Mitsuyo Okazaki[5], Dongfang Li[2], Noboru Mizushima[1*]

[1]Department of Biochemistry and Molecular Biology, Graduate School and Faculty of Medicine, University of Tokyo, Tokyo, Japan; [2]National Laboratory of Biomacromolecules, CAS Center for Excellence in Biomacromolecules, Institute of Biophysics, Chinese Academy of Sciences, Beijing, China; [3]Department of Molecular, Cell and Cancer Biology, University of Massachusetts Medical School, Worcester, United States; [4]Microscopy Research Support Unit Research Core, Tokyo Medical and Dental University, Tokyo, Japan; [5]Tokyo Medical and Dental University, Tokyo, Japan

*For correspondence:
nmizu@m.u-tokyo.ac.jp

[†]These authors contributed equally to this work

Present address: [‡]Department of Physiology, Juntendo University Graduate School of Medicine, Tokyo, Japan; [§]Department of Biochemistry and Cell Biology, National Institute of Infectious Diseases, Tokyo, Japan; [#]Molecular Cell Biology of Autophagy, The Francis Crick Institute, London, United Kingdom

**Abstract** Lipoproteins are lipid-protein complexes that are primarily generated and secreted from the intestine, liver, and visceral endoderm and delivered to peripheral tissues. Lipoproteins, which are assembled in the endoplasmic reticulum (ER) membrane, are released into the ER lumen for secretion, but its mechanism remains largely unknown. Here, we show that the release of lipoproteins from the ER membrane requires VMP1, an ER transmembrane protein essential for autophagy and certain types of secretion. Loss of *vmp1*, but not other *autophagy-related genes,* in zebrafish causes lipoprotein accumulation in the intestine and liver. *Vmp1* deficiency in mice also leads to lipid accumulation in the visceral endoderm and intestine. In VMP1-depleted cells, neutral lipids accumulate within lipid bilayers of the ER membrane, thus affecting lipoprotein secretion. These results suggest that VMP1 is important for the release of lipoproteins from the ER membrane to the ER lumen in addition to its previously known functions.
DOI: https://doi.org/10.7554/eLife.48834.001

## Introduction

Lipoproteins are lipid-protein complexes whose main function is to transport hydrophobic lipids derived from dietary and endogenous fat to peripheral tissues by the circulation systems for energy utilization or storage. Lipoproteins are primarily formed in and secreted from the intestine, liver, and visceral endoderm (*Farese et al., 1996*; *Sirwi and Hussain, 2018*). Lipoproteins are composed of a neutral lipid core (triglycerides and cholesterol esters) surrounded by a phospholipid monolayer and proteins (called apolipoproteins). At an early stage in lipoprotein assembly, neutral lipids are synthesized and accumulate within the lipid bilayer of the endoplasmic reticulum (ER) membrane (*Demignot et al., 2014*; *Sundaram and Yao, 2010*; *Tiwari and Siddiqi, 2012*; *Yen et al., 2015*). These lipid structures are associated with apolipoprotein B (APOB), a major protein constituent of lipoproteins, co-and/or post-translationally (*Davidson and Shelness, 2000*). This step requires microsomal triglyceride-transfer protein (MTTP), an ER luminal chaperone that interacts, stabilizes, and lipidates APOB (*Sirwi and Hussain, 2018*). Then, lipoproteins are released into the ER lumen (*Demignot et al., 2014*; *Sundaram and Yao, 2010*; *Tiwari and Siddiqi, 2012*; *Yen et al., 2015*) and transported to the Golgi for secretion. A key long-standing question is how lipoproteins bud off from the ER membrane to the ER lumen, which remains largely unknown.

Vacuole membrane protein 1 (VMP1), which was originally identified as a pancreatitis-associated protein, is a multispanning membrane protein in the ER (*Dusetti et al., 2002*; *Vaccaro et al., 2003*). VMP1 (EPG-3 in *Caenorhabditis elegans*) is essential for autophagosome formation in mammals (*Itakura and Mizushima, 2010*; *Ropolo et al., 2007*; *Tian et al., 2010*), *Dictyostelium* (*Calvo-Garrido et al., 2008*), and *Caenorhabditis elegans* (*Tian et al., 2010*). Although VMP1 may regulate the PI3K complex I signal (*Calvo-Garrido et al., 2014*; *Kang et al., 2011*; *Ropolo et al., 2007*), which is required for autophagy (*Ktistakis and Tooze, 2016*; *Mizushima et al., 2011*; *Nakatogawa et al., 2009*; *Søreng et al., 2018*), VMP1 also controls ER contact with other membranes, including autophagic membranes (*Tábara and Escalante, 2016*; *Zhao et al., 2017*), by regulating the calcium pump sarcoendoplasmic reticulum calcium transport ATPase (SERCA) (*Zhao et al., 2017*) and ER contact proteins VAPA and VAPB (*Zhao et al., 2018*). At the ER-autophagic membrane contact sites, VMP1 forms ER subdomains enriched in phosphatidylinositol synthase (*Tábara et al., 2018*), which could serve as the initiation site of autophagosome formation (*Nishimura et al., 2017*).

In addition to the involvement in autophagy, VMP1 is known to be required for the secretion of soluble proteins that are transported via the ER-to-Golgi trafficking pathway. In *Drosophila* S2 cells, VMP1 (identified as TANGO5) is important for constitutive secretion and Golgi organization (*Bard et al., 2006*). In *Dictyostelium*, VMP1 is required for secretion of specific proteins such as α-mannosidase and a cysteine proteinase and maintenance of organelle homeostasis (*Calvo-Garrido et al., 2008*).

Physiologically, VMP1 is essential for survival under hypoosmotic and starvation conditions in *Dictyostelium* (*Calvo-Garrido et al., 2008*) and *Caenorhabditis elegans* (*Tian et al., 2010*), respectively. However, its physiological roles in vertebrates remain unknown. Recent studies in human cells (*Morita et al., 2018*; *Tábara and Escalante, 2016*; *Zhao et al., 2017*) and *Caenorhabditis elegans* (*Zhao et al., 2017*) revealed that neutral lipid-containing structures accumulate in VMP1-depleted cells, suggesting the function of VMP1 in lipid metabolism. In this study, via deletion of the *VMP1* gene, we found that VMP1 is essential for survival during the larval and early embryonic periods in zebrafish and mice, respectively. We further revealed that VMP1 is important for lipoprotein release from the ER membrane into the lumen to be secreted from the intestine, liver, and visceral endoderm. This function is distinct from previously known functions of VMP1 in autophagy and secretion.

## Results

### Loss of *vmp1* in zebrafish causes larval lethality and defects in autophagy

To reveal the physiological functions of VMP1 in vertebrates, we used zebrafish and mice. We generated *vmp1*-deficient zebrafish using the CRISPR/Cas9 system. A frameshift mutation was introduced into exon 6 of the *vmp1* gene (*Figure 1A*). Gross examination revealed that the abdominal part was less transparent in *vmp1*$^{-/-}$ zebrafish at 6 days post fertilization (dpf), indicating the presence of abnormal deposits (*Figure 1B*). We also noticed that the swimbladder was not inflated in *vmp1*$^{-/-}$ zebrafish, which will be described in more detail elsewhere. All *vmp1*$^{-/-}$ zebrafish died around at nine dpf (*Figure 1C*), suggesting that VMP1 is essential for survival during the larval period.

Autophagy was defective in *vmp1*$^{-/-}$ zebrafish; many large LC3 puncta accumulated in several tissues, including the brain, spinal cord, and skeletal muscles, which were abnormal autophagy-related structures typically observed in VMP1-deficient mammalian cells (*Itakura and Mizushima, 2010*; *Kishi-Itakura et al., 2014*; *Zhao et al., 2017*) (*Figure 1D*). An increase in the levels of the lipidated form of LC3 (LC3-II) was also observed in *vmp1*$^{-/-}$ zebrafish (*Figure 1E*), as previously observed in VMP1-deficient mammalian cells (*Itakura and Mizushima, 2010*; *Morita et al., 2018*; *Shoemaker et al., 2019*; *Zhao et al., 2017*). These results suggest that autophagic flux is blocked in *vmp1*$^{-/-}$ zebrafish.

### Accumulation of neutral lipids in intestinal epithelial cells and hepatocytes in *vmp1*-deficient zebrafish

The abnormal deposits in the abdomen were observed in all *vmp1*$^{-/-}$ zebrafish (n = 11), but not in *vmp1*$^{+/-}$ (n = 30) or *vmp1*$^{+/+}$ zebrafish (n = 7). These deposits resemble neutral lipid accumulation in

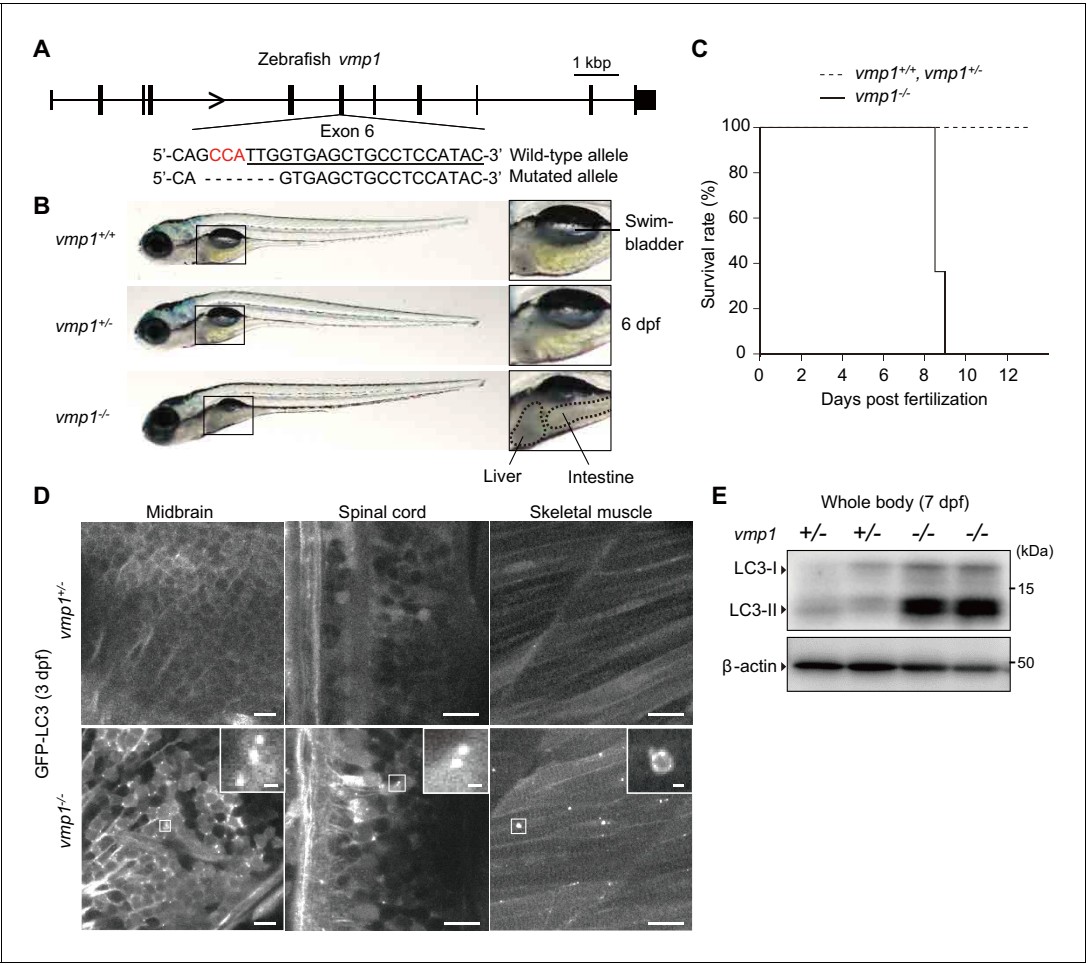

**Figure 1.** Loss of *vmp1* in zebrafish causes lethality around 9 days post fertilization and defective autophagy. (**A**) Schematic representation of the Cas9-gRNA-targeted site in the zebrafish *vmp1* genomic locus. The protospacer-adjacent motif (PAM) sequence is shown in red. The targeted site is underlined. A 7 bp deletion in the mutated allele is shown. (**B**) External appearance of 6-dpf *vmp1*[+/+], *vmp1*[+/-], and *vmp1*[-/-] zebrafish. Magnified images of the indicated regions are shown in the right panels. Dashed lines indicate abnormal deposits in the liver and intestine. Data are representative of four independent experiments. (**C**) Survival rate (% of total fish) of *vmp1*[+/+] (n = 7), *vmp1*[+/-] (n = 30), and *vmp1*[-/-] (n = 11) zebrafish. Data are representative of two independent experiments. (**D**) Representative images of GFP-LC3 signals in the midbrain, spinal cord, and skeletal muscle of 3-dpf *vmp1*[+/-] and *vmp1*[-/-] zebrafish injected with GFP-LC3 mRNA. Data are representative of two independent experiments. Scale bars, 10 μm and 1 μm in the inset. (**E**) Immunoblotting of LC3 and β-actin in two 7-dpf *vmp1*[+/-] and *vmp1*[-/-] zebrafish. Data are representative of two independent experiments.

DOI: https://doi.org/10.7554/eLife.48834.002

The following source data is available for figure 1:

**Source data 1.** Related to *Figure 1C*.

DOI: https://doi.org/10.7554/eLife.48834.003

the intestine (*Hölttä-Vuori et al., 2010*). Indeed, the deposits in *vmp1*[-/-] zebrafish were stained with oil red O, a neutral lipid-soluble dye (*Figure 2A*). Oil red O staining of cross sections and electron microscopy revealed that, in *vmp1*[-/-] zebrafish, large neutral lipid-containing structures accumulated in intestinal epithelial cells and hepatocytes (*Figure 2B,C*) but not in other organs, including the brain (*Figure 2—figure supplement 1A,B*) and skeletal muscles (*Figure 2—figure supplement 1A, C*). Accumulation of large lipid-containing structures was not observed in zebrafish lacking the *rb1cc1/fip200* (*Figure 2—figure supplement 1D*) or *atg5* (*Figure 2—figure supplement 1E*) gene, both of which are required for autophagy (*Hara et al., 2008*; *Mizushima et al., 2001*). These results suggest that neutral lipids accumulate in *vmp1*[-/-] zebrafish, and that this lipid phenotype is not caused by deficient autophagy.

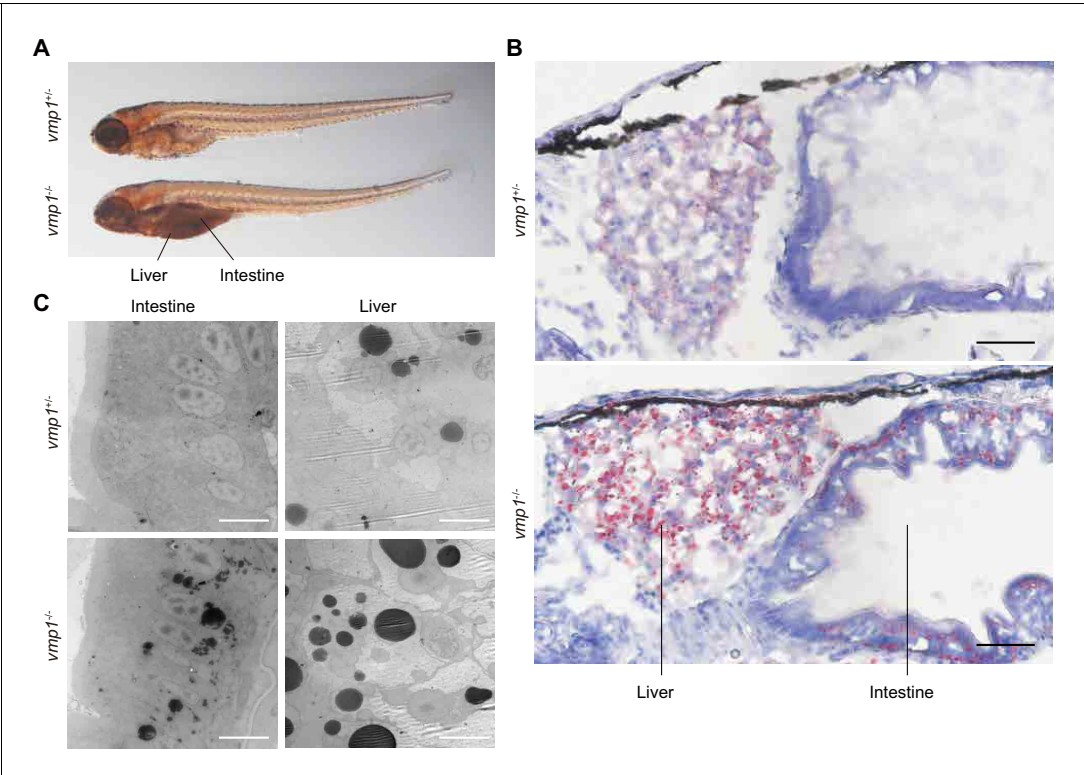

**Figure 2.** Loss of *vmp1* in zebrafish causes accumulation of neutral lipids in the intestine and liver. (A) Whole-mount oil red O staining of 8.5-dpf *vmp1*$^{+/-}$ and *vmp1*$^{-/-}$ zebrafish. Data are representative of three independent experiments. (B) Oil red O and hematoxylin staining of 6-dpf *vmp1*$^{+/-}$ and *vmp1*$^{-/-}$ zebrafish. Data are representative of two independent experiments. Scale bars, 20 μm. (C) Transmission electron microscopy of the intestine and liver from 6-dpf *vmp1*$^{+/-}$ and *vmp1*$^{-/-}$ zebrafish. Data are representative of three independent experiments. Scale bars, 5 μm.
DOI: https://doi.org/10.7554/eLife.48834.004

The following figure supplement is available for figure 2:

**Figure supplement 1.** Large lipid-containing structures are not observed in the brain and skeletal muscle of *vmp1*$^{-/-}$ zebrafish or in the intestine of *rb1cc1*$^{-/-}$ and *atg5*$^{-/-}$ zebrafish.
DOI: https://doi.org/10.7554/eLife.48834.005

## Loss of *Vmp1* in mice causes early embryonic lethality and accumulation of lipids in visceral endoderm cells

To elucidate the physiological functions of VMP1 in mammals, *Vmp1*-deficient mice were generated using an embryonic stem (ES) cell line carrying a gene-trap cassette downstream of exon 3 of the *Vmp1* gene (*Figure 3—figure supplement 1A*). Heterozygous *Vmp1*$^{gt/+}$ mice were healthy and phenotypically indistinguishable from wild-type littermates. In contrast, *Vmp1*$^{gt/gt}$ embryos were embryonic lethal; they were detected at 7.5 days postcoitum (dpc) but not after 9.5 dpc (*Figure 3A*). *Vmp1*$^{gt/gt}$ embryos at 7.5 dpc were smaller than wild-type embryos and accumulated the autophagy substrate p62 (*Figure 3B*), suggesting that VMP1 is important for early embryonic development as well as autophagy in mice.

The visceral endoderm is an extraembryonic layer critical for maternal-to-embryo transfer of nutrients such as neutral lipids between 5 and 10 dpc, before the placenta is formed (*Bielinska et al., 1999*). Like intestinal epithelial cells and hepatocytes, visceral endoderm cells secrete lipoproteins to the epiblast, an embryonic layer (*Farese et al., 1996*). Thus, we examined lipid distribution in these embryos. Indeed, neutral lipids accumulated in visceral endoderm cells in *Vmp1*$^{gt/gt}$ embryos at 7.5 dpc (*Figure 3C*), as observed in *vmp1*-deficient zebrafish intestinal epithelial cells and hepatocytes.

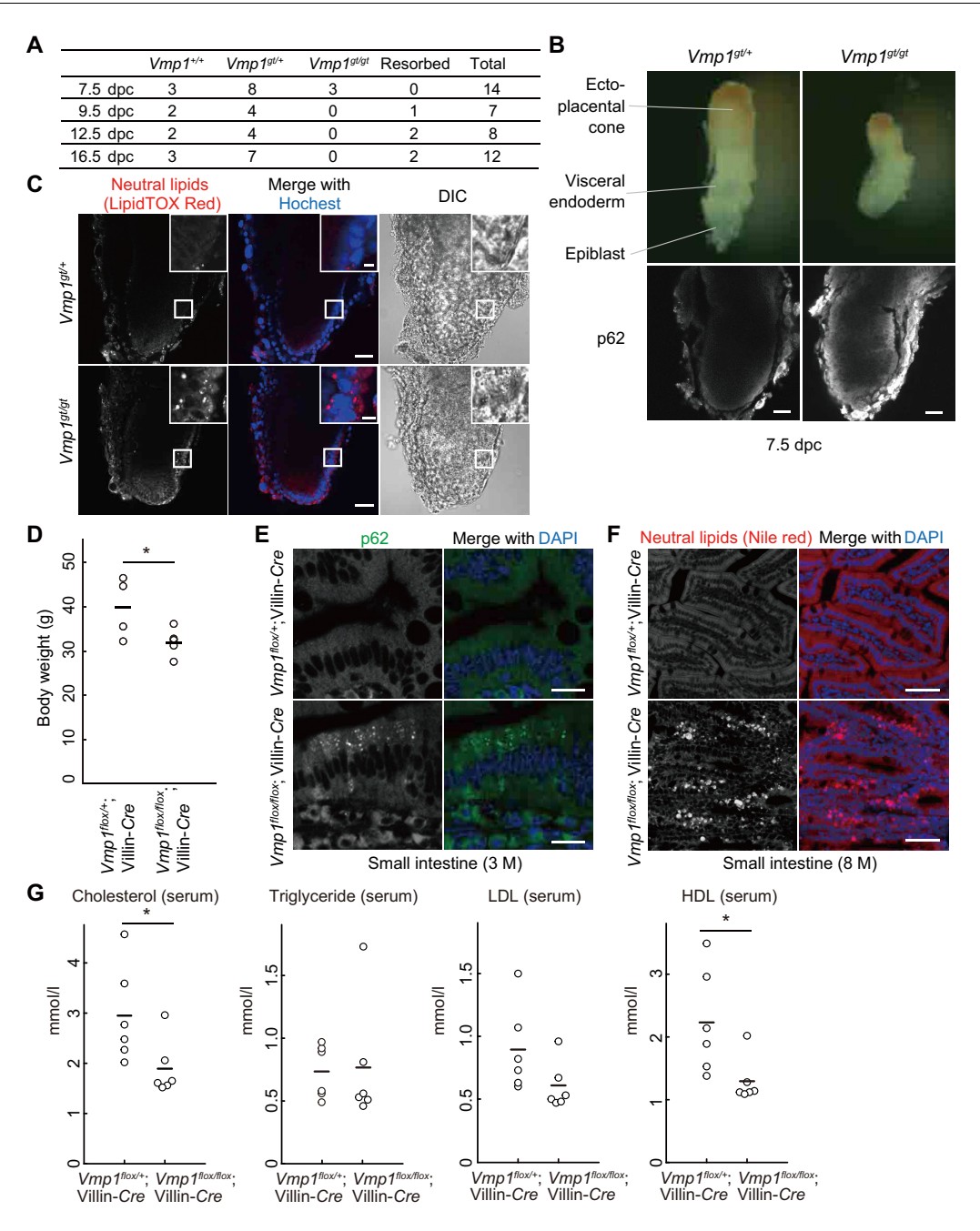

**Figure 3.** Systemic and intestinal epithelial cell-specific deletion of *Vmp1* in mice causes accumulation of neutral lipids. (A) Genotypes of offspring from *Vmp1*<sup>gt/+</sup> intercross. (B) 7.5-dpc embryos were extracted from the conceptus and stained with anti-p62 antibody. Data are representative of two independent experiments. Scale bars, 50 μm. (C) 7.5-dpc embryos were stained with LipidTOX Red and Hoechst33342. The visceral endoderm cells are magnified in the insets. Data are representative of two independent experiments. Scale bars, 50 μm and 10 μm in the insets. (D) Body weight of *Vmp1*<sup>flox/+</sup>;Villin-*Cre* (n = 4) and *Vmp1*<sup>flox/flox</sup>;Villin-*Cre* (n = 5) male mice at 7–10 months of age. The horizontal lines indicate the means for each group. Differences were determined by unpaired Student *t*-test (*, p<0.05). (E) The small intestine from 3-month-old *Vmp1*<sup>flox/+</sup>;Villin-*Cre* and *Vmp1*<sup>flox/flox</sup>; Villin-*Cre* mice was stained with anti-p62 antibody and DAPI. Scale bars, 20 μm. (F) The small intestine from 8-month-old *Vmp1*<sup>flox/+</sup>;Villin-*Cre* and *Vmp1*<sup>flox/flox</sup>;Villin-*Cre* mice fed *ad libitum* was stained with Nile red and DAPI. Scale bars, 50 μm. (G) The amount of serum cholesterol, triglyceride, LDL, and HDL in 18-month-old *Vmp1*<sup>flox/+</sup>;Villin-*Cre* and *Vmp1*<sup>flox/flox</sup>;Villin-*Cre* mice fed *ad libitum*. The horizontal lines indicate the means for each group. Differences were determined by unpaired Student *t*-test (*, p<0.05). LDL, low-density lipoprotein; HDL, high-density lipoprotein.

DOI: https://doi.org/10.7554/eLife.48834.006

The following source data and figure supplement are available for figure 3:

*Figure 3 continued on next page*

*Figure 3 continued*

**Source data 1.** Related to *Figure 3D,G*.
DOI: https://doi.org/10.7554/eLife.48834.008
**Figure supplement 1.** Genetic map of the gene-trap and floxed alleles of the mouse *Vmp1* gene in *Vmp1^gt^* mice and *Vmp1^flox^* mice, respectively.
DOI: https://doi.org/10.7554/eLife.48834.007

## Intestinal epithelial cell-specific loss of *Vmp1* in mice causes accumulation of lipids in intestinal epithelial cells

To circumvent the lethality of *Vmp1^gt/gt^* mouse embryos and study the role of VMP1 in the intestine, we generated intestinal epithelial cell-specific *Vmp1*-deficient mice. Mice harboring a *Vmp1^flox^* allele were crossed with Villin-*Cre* transgenic mice expressing Cre recombinase under the control of the Villin promoter (*el Marjou et al., 2004*) (*Figure 3—figure supplement 1B*). *Vmp1^flox/flox^*;Villin-*Cre* mice weighed less than *Vmp1^flox/+^*;Villin-*Cre* mice (*Figure 3D*). Accumulation of p62 was observed in intestinal epithelial cells in 3-month-old *Vmp1^flox/flox^*;Villin-*Cre* mice (*Figure 3E*). The intestines of 8-month-old *Vmp1^flox/flox^*;Villin-*Cre* mice showed accumulation of neutral lipids within intestinal epithelial cells (*Figure 3F*), suggesting a conserved function of VMP1 in the intestine. In the serum from *Vmp1^flox/flox^*;Villin-*Cre* mice, the levels of cholesterol and lipoproteins such as high density lipoprotein (HDL) decreased compared to those from *Vmp1^flox/+^*;Villin-*Cre* mice (*Figure 3G*). These results suggest that VMP1 is critical for homeostasis of neutral lipids and lipoproteins in a whole body.

## VMP1 is important for secretion of lipoproteins

Next, we examined the mechanism by which neutral lipids accumulate in VMP1-depleted organisms. Because intestinal epithelial cells, hepatocytes, and visceral endoderm cells are active in the secretion of lipoproteins (*Farese et al., 1996*; *Sirwi and Hussain, 2018*), we hypothesized that a block in lipoprotein secretion is the cause of lipid accumulation. To this end, we used the human hepatocellular carcinoma cell line HepG2 because these cells constitutively secrete lipoproteins. In *VMP1*-silenced HepG2 cells, the amount of triglyceride and cholesterol decreased in culture media (*Figure 4A,B*). In contrast, the intracellular amounts increased (*Figure 4A,B*), suggesting that the secretion of neutral lipids depends on VMP1. Chromatographic analysis using different detection methods for neutral lipids also revealed significant reductions in lipoproteins such as very low-density lipoproteins and low-density lipoproteins in the culture media of *VMP1*-silenced HepG2 cells both under normal and oleic acid-treated conditions, the latter of which stimulates lipoprotein secretion (*Figure 4—figure supplement 1A,B*). Likewise, knockdown of VMP1 reduced the amount of APOB in culture media (*Figure 4C*). Paradoxically, it also reduced the amount of intracellular APOB (*Figure 4C*). This was likely due to enhanced degradation of misfolded APOB by the ubiquitin-proteasome system because treatment with lactacystin, a proteasome inhibitor, restored the amount of intracellular APOB (*Figure 4—figure supplement 1C*). Nevertheless, the amount of extracellular APOB remained lower in *VMP1*-silenced cells than that in control cells (*Figure 4—figure supplement 1C*). These results suggest that VMP1 is critical not only for APOB homeostasis but also for the secretion of lipoproteins.

We next investigated whether VMP1 is required for secretion of proteins besides APOB. Secretion of APOE, a component of very low density lipoprotein (VLDL), and APOA-I, a component of HDL, was also impaired in *VMP1*-silenced HepG2 cells (*Figure 4C*). Secretion of APOA-I was affected only slightly. In contrast, secretion of albumin, which is transported from the ER to Golgi separately from lipoproteins (*Tiwari and Siddiqi, 2012*), was not significantly impaired in *VMP1*-silenced HepG2 cells (*Figure 4C*). Secretion of collagens, another type of large cargo that requires TANGO1 for secretion (*Saito et al., 2009*), was not affected by VMP1 deletion because cartilage structures, which are composed of collagens secreted from cartilage cells, was normal in *vmp1^-/-^* zebrafish (*Figure 4—figure supplement 1D,E*). This result is consistent with a previous report that showed the secretion of collagens and model cargo proteins such as VSVG, a glycoprotein of vesicular stomatitis virus, is normal in *vmp1/epg-3* mutant *Caenorhabditis elegans* and *VMP1* knockout COS7 cells, respectively (*Zhao et al., 2017*). Thus, a defect in secretion in VMP1-deficient cells is not general, but rather specific to lipoproteins.

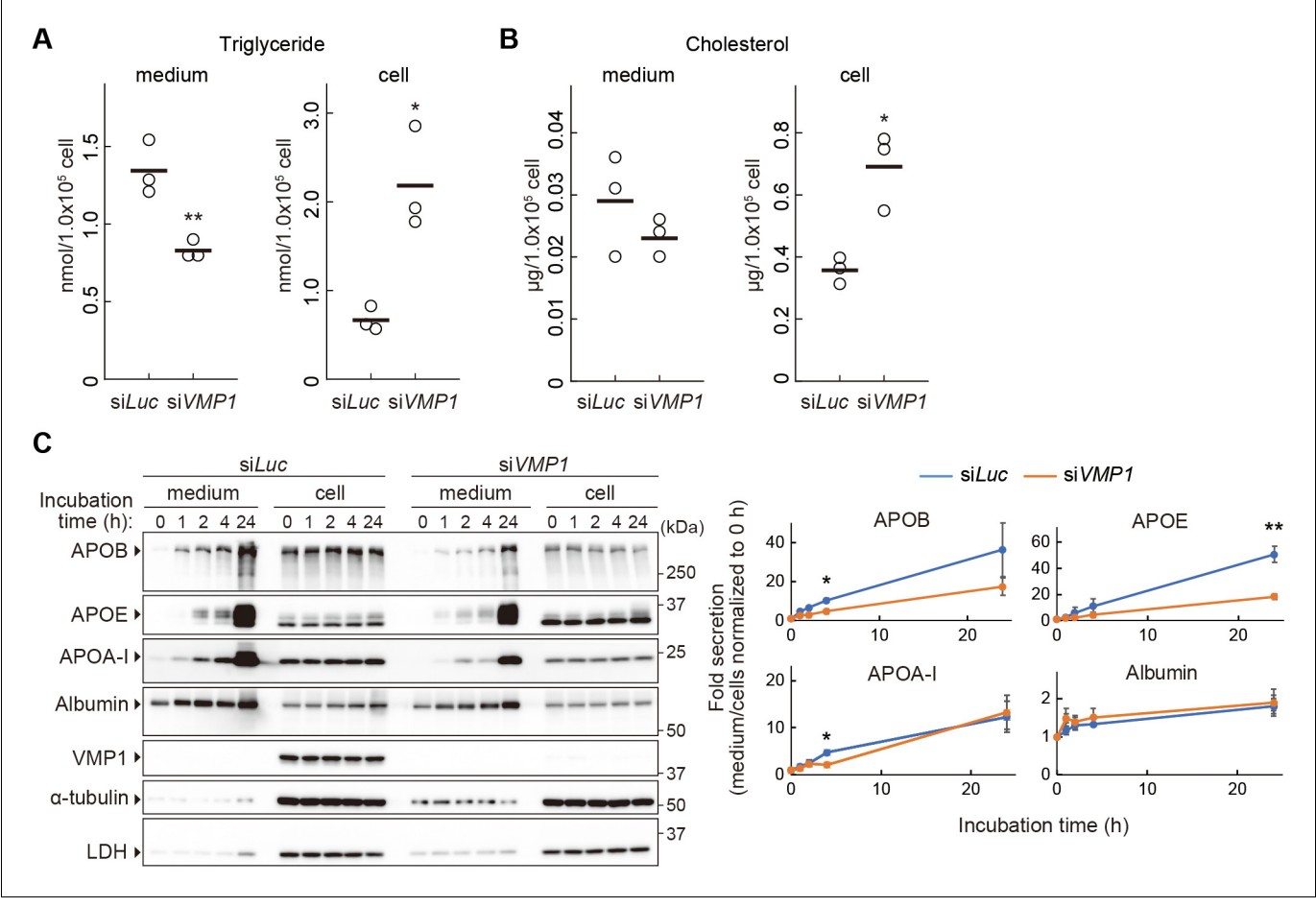

**Figure 4.** VMP1 is important for secretion of lipoproteins. (**A and B**) HepG2 cells were treated with siRNA against *luciferase* (*Luc*) or *VMP1* and cultured in serum-free medium for 24 hr. Triglycerides (**A**) and cholesterols (**B**) were extracted from culture medium and cells, measured and analyzed using the Student's *t*-test (\*\*, p<0.01; \*, p<0.05). The horizontal lines indicate the means of three independent experiments for each group. (**C**) HepG2 cells were treated as in (**A**) and cultured in regular medium containing 200 nM oleic acid for 24 hr. Cells were then washed and re-cultured in serum-free medium for indicated times. The medium was concentrated by TCA precipitation. Samples (approximately 7% or 14% vol of total precipitated media or cell lysates, respectively) were subjected to immunoblot analysis. The amount of proteins was quantified through densitometric scanning of band intensities and the medium/cells ratio was determined. Data represent the mean ± standard error of the mean (n = 3), which was normalized to 0 hr, and statistically analyzed using the Student's *t*-test (\*\*, p<0.01; \*, p<0.05).

DOI: https://doi.org/10.7554/eLife.48834.009

The following source data and figure supplements are available for figure 4:

**Source data 1.** Related to *Figure 4A–C*.
DOI: https://doi.org/10.7554/eLife.48834.012

**Figure supplement 1.** VMP1 is required for secretion and homeostasis of lipoproteins but not for formation of cartilage structures in the zebrafish head skeleton.
DOI: https://doi.org/10.7554/eLife.48834.010

**Figure supplement 1—source data 1.** Related to *Figure 4—figure supplement 1A–C*.
DOI: https://doi.org/10.7554/eLife.48834.011

## Neutral lipids accumulate in the ER in the absence of VMP1

In intestinal epithelial cells and hepatocytes, neutral lipids are synthesized within the lipid bilayer of the ER membrane and released into the ER lumen for secretion (*Demignot et al., 2014*; *Sundaram and Yao, 2010*; *Tiwari and Siddiqi, 2012*; *Yen et al., 2015*). In *vmp1*[-/-] zebrafish, almost all large lipid-containing structures in intestinal epithelial cells and hepatocytes were surrounded by the ER transmembrane protein Sec61B (*Figure 5A*). In most cases, Sec61B covered only a part rather than all of the surface of the lipid structures. Also, in *Vmp1*[gt/gt] mouse embryos, almost all neutral

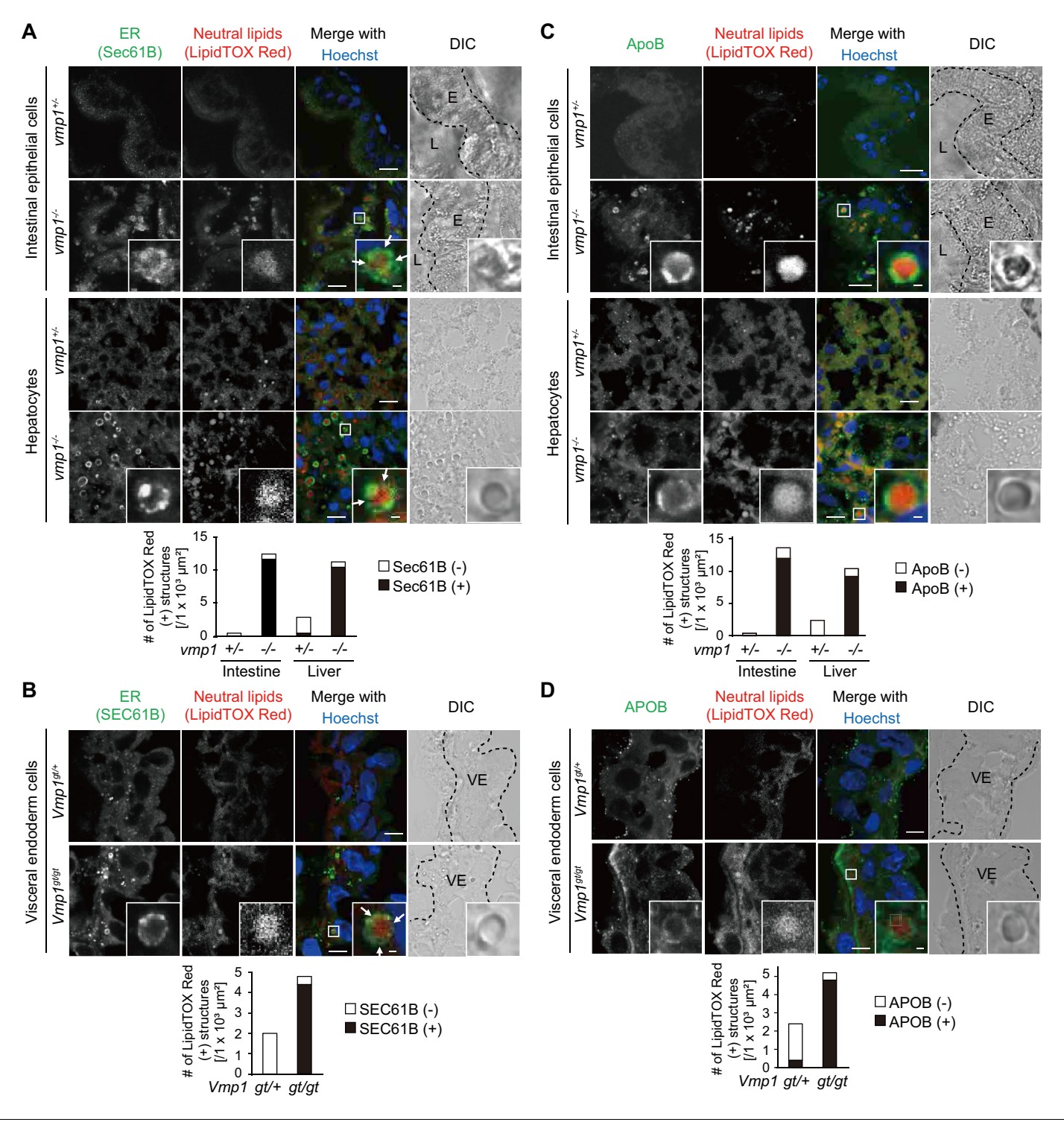

**Figure 5.** *Vmp1*-deficient zebrafish and mice show accumulation of lipoproteins in the intestine, liver, and visceral endoderm. Immunohistochemistry of the intestine and liver from 6-dpf *vmp1^+/-^* and *vmp1^-/-^* zebrafish (**A** and **C**) and the visceral endoderm from 7.5-dpc *Vmp1^gt/+^* and *Vmp1^gt/gt^* mice (**B** and **D**) using anti-SEC61B antibody (**A** and **B**), anti-APOB antibody (**C** and **D**), LipidTOX Red, and Hoechst33342. Arrows indicate the regions where the Sec61B/SEC61B signals were weak. The regions of zebrafish intestinal epithelial cells (E), intestinal lumen (L) or mouse visceral endoderm cells (VE) are shown as dashed lines. Data are representative of two independent experiments. Scale bars, 10 μm and 1 μm in the inset. The number of LipidTOX Red (+) structures with (black columns) or without (white columns) SEC61B (**A and B**) or APOB (**C and D**) per observed area was analyzed from at least two randomly selected areas using ImageJ software.

*Figure 5 continued on next page*

*Figure 5 continued*

DOI: https://doi.org/10.7554/eLife.48834.013

The following source data is available for figure 5:

**Source data 1.** Related to *Figure 5A–D*.

DOI: https://doi.org/10.7554/eLife.48834.014

lipid-containing structures were positive for SEC61B (*Figure 5B*). In contrast, neutral lipid structures in these tissues in *vmp1*$^{+/-}$ animals were mostly negative for Sec61B/SEC61B, suggesting that they are present outside the ER, most likely as cytosolic lipid droplets (*Figure 5A,B*). Thus, neutral lipids abnormally accumulate in the ER in *vmp1*-deficient zebrafish and mouse tissues.

## VMP1 is important for the release of lipoproteins from the ER membrane

We further narrowed down the step defective in VMP1-deficient cells. Neutral lipids accumulating within lipid bilayers of the ER are released into the ER lumen to form lipoproteins together with APOB (*Sirwi and Hussain, 2018*). In *vmp1*$^{-/-}$ zebrafish, most of the lipid-containing structures were positive for ApoB (*Figure 5C*). In addition, the lipid structures were mostly positive for APOB in *Vmp1*$^{gt/gt}$ mouse embryos (*Figure 5D*). In agreement with SEC61B staining data, most lipid structures in these tissues in *vmp1*$^{+/-}$ animals were ApoB/APOB-negative (*Figure 5C,D*). These results suggest that lipoproteins or lipoprotein-related structures are formed and accumulate in VMP1-deficient cells.

In wild-type HepG2 cells, neutral lipid structures were mostly positive for adipose differentiation-related protein (ADRP, also known as perilipin 2), a marker for cytosolic lipid droplets, but negative for APOB irrespective of oleic acid treatment that increased the number of lipid-containing structures (*Figure 6A–C*), suggesting that these are lipid droplets rather than lipoproteins. In contrast, as shown in zebrafish and mice (*Figure 5*), large lipid structures accumulated in *VMP1*-silenced HepG2 cells (*Figure 6A*) and most of them were APOB positive (*Figure 6C*). Some of them were positive for both APOB and ADRP, where APOB and ADRP were distributed into distinct regions (*Figure 6D*). They should represent structures stuck within the ER lipid bilayers facing both the cytosol and the ER lumen, rather than those released into the ER lumen (*Figure 6D*). APOE, but not APOA-I, colocalized with APOB on the lipid structures in *VMP1*-silenced HepG2 cells (*Figure 6E,F*), suggesting that the defective secretion of APOB and APOE (*Figure 4C*) is at least partly caused by trapping in the lipid structures.

Similar crescent-shaped accumulations of APOB and ADRP around lipids trapped within the ER membranes were also observed in human hepatoma cell line Huh7 cells treated with proteasome inhibitors (*Ohsaki et al., 2008*). In *VMP1*-silenced HepG2 cells, however, proteasome activity was not suppressed (*Figure 6—figure supplement 1A*). In addition, treatment of wild-type HepG2 cells with proteasome inhibitors (MG132 or lactacystin) did not induce crescent-shaped accumulations of APOB and ADRP (*Figure 6—figure supplement 2A*). These results are somehow different from those in the previous report (*Ohsaki et al., 2008*), probably because of a difference in cell types or culture conditions. The crescent-shaped accumulations of APOB and ADRP was also observed by treatment with docosahexaenoic acid or cyclosporin A (*Ohsaki et al., 2008*), which induce APOB proteolysis by unknown molecular mechanisms (*Fisher et al., 2001*; *Kaptein et al., 1994*), suggesting the possible involvement of APOB proteolysis in the formation of these structures. APOB was degraded by induction of ER stress or depletion of MTTP (*Ota et al., 2008*; *Sirwi and Hussain, 2018*). However, neither ER stress (*Figure 6—figure supplement 1B*) or reduced MTTP protein level (*Figure 6—figure supplement 1C*) was observed in *VMP1*-silenced HepG2 cells. Treatment of wild-type HepG2 cells with ER stress inducers (tunicamycin or thapsigargin) (*Figure 6—figure supplement 2B*) or an MTTP inhibitor (CP-346086) (*Figure 6—figure supplement 2C*) did not induce the crescent-shaped accumulations of APOB and ADRP. Furthermore, the crescent-shaped accumulations of APOB and ADRP were not observed in HepG2 cells deficient for FITM2 (*Figure 6—figure supplement 2D,E*), a factor required for budding of lipid droplets from the ER membrane to the cytosol, but not for lipoprotein secretion (*Choudhary et al., 2015*; *Goh et al., 2015*; *Kadereit et al., 2008*). Taken together, these results suggest that the crescent-shaped

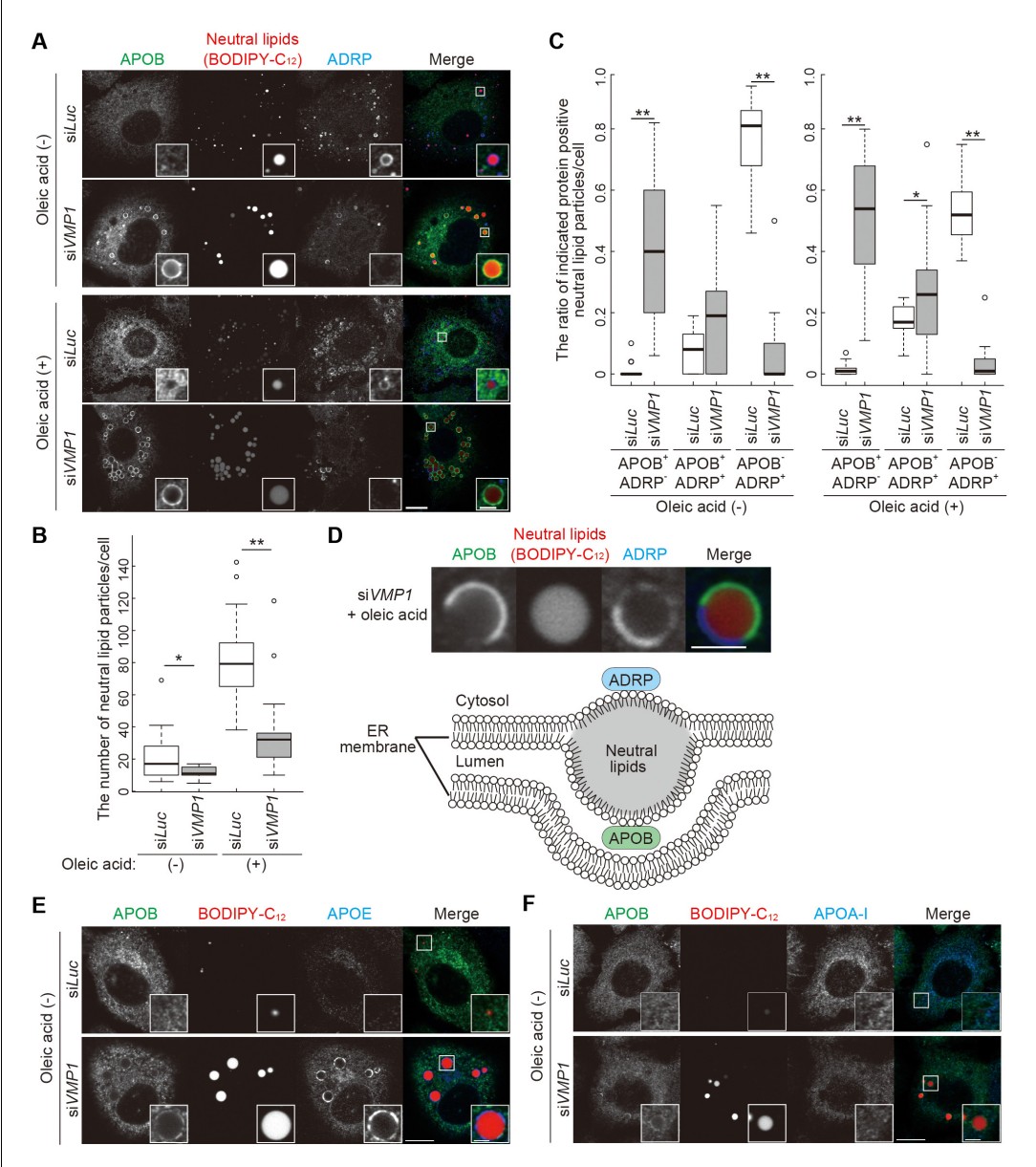

**Figure 6.** Depletion of VMP1 in HepG2 cells causes accumulation of abnormal lipoproteins. (A–C) HepG2 cells were treated with siRNA oligonucleotides against *luciferase* (*Luc*) or *VMP1*, cultured in regular medium in the presence or absence of 200 nM oleic acid for 24 hr, and stained with BODIPY-$C_{12}$ 558/568 for 1 hr to visualize the neutral lipids. Cells were fixed and stained with anti-APOB and anti-ADRP antibodies. Scale bars, 10 μm and 2 μm in the inset. The number of neutral lipid particles per cell (B) and ratio of APOB- or ADRP-positive neutral lipid particles (C) was quantified. Solid bars indicate median, boxes the interquartile range (25th to 75th percentile), and whiskers 1.5 times the interquartile range. The outliers are plotted individually. Differences were determined by Mann-Whitney U-test (\*\*, p<0.01; \*, p<0.05; n ≥ 17 cells). (D) Representative images of APOB- and ADRP-double positive neutral lipid particles in VMP1-depleted HepG2 cells. Scale bars, 2 μm. A model of APOB- and ADRP-double positive neutral lipid particles in VMP1-depleted cells is shown. (E and F) HepG2 cells were treated as in (A), cultured in regular medium, and stained with BODIPY-$C_{12}$ 558/568 for 1 hr. Cells were fixed and stained with indicated antibodies. Scale bars,10 μm and 2 μm in the inset.

DOI: https://doi.org/10.7554/eLife.48834.015

The following source data and figure supplements are available for figure 6:

**Source data 1.** Related to *Figure 6B,C*.
DOI: https://doi.org/10.7554/eLife.48834.020
**Figure supplement 1.** Depletion of VMP1 in HepG2 cells does not affect proteasome activity, ER stress, and MTTP expression.
DOI: https://doi.org/10.7554/eLife.48834.016
**Figure supplement 1—source data 1.** Related to *Figure 6—figure supplement 1A*.

*Figure 6 continued on next page*

*Figure 6 continued*

DOI: https://doi.org/10.7554/eLife.48834.017

**Figure supplement 2.** APOB- and ADRP-double positive structures are not formed by proteasome inhibition, ER stress induction, MTTP inhibition, or depletion of FITM2.

DOI: https://doi.org/10.7554/eLife.48834.018

**Figure supplement 2—source data 1.** Related to *Figure 6—figure supplement 2A,D,E*.

DOI: https://doi.org/10.7554/eLife.48834.019

accumulations of APOB and ADRP in *VMP1*-silenced HepG2 cells is not due to proteasome inhibition, ER stress, MTTP suppression, or defective budding of lipid droplets from the ER membrane.

Electron microscopy of intestinal epithelial cells (*Figure 7A*) and hepatocytes (*Figure 7B*) of *vmp1*$^{-/-}$ zebrafish and *VMP1*-silenced HepG2 cells (*Figure 7C*) revealed that the ER membranes with ribosomes on their cytosolic face covered a part of the surface of the lipid structures (*Figure 7A–C*, black arrowheads) and fused with the lipid structures at both ends (*Figure 7A–C*, white arrowheads). The space between the ER membrane and the lipid structures should be the ER lumen (*Figure 7D*). In contrast, lipid accumulation within the ER membrane was not observed in *vmp1*$^{+/-}$ zebrafish and wild-type HepG2 cells (*Figure 7A–C*, arrows). In hepatocytes in *vmp1*$^{+/-}$ zebrafish, there was no membrane on large lipid structures, which should represent cytosolic lipid droplets (*Figure 7B*). These results suggest that neutral lipids abnormally accumulate within the lipid bilayer of the ER membrane in the absence of VMP1, and VMP1 is important for the release of lipoproteins into the ER lumen (*Figure 7D*).

## Discussion

Based on the findings in this study, we propose that the ER protein VMP1 has a novel nonautophagic function in the release of lipoproteins from the ER membrane into the lumen (*Figure 7D*). This step is distinct from the exit from the ER; it is generally thought that intraluminal lipoproteins are transported to the Golgi (*Figure 7D*). Consistently, the phenotype of deletion of VMP1 is different from that of factors required for lipoprotein export from the ER lumen to the Golgi (*Figure 7D*); deletion of TANGO1, TALI (*Santos et al., 2016*), cTAGE5 (*Wang et al., 2016*), or SURF4 (*Saegusa et al., 2018*) does not cause an accumulation of large lipid-containing structures in human hepatocellular carcinoma cells or epithelial colorectal adenocarcinoma cells. However, as it is technically difficult to definitely demonstrate where each apolipoprotein and neutral lipids accumulate in the ER, we do not exclude the possibility that VMP1 is also important at the step of ER-to-Golgi budding, which is not mutually exclusive.

Our results also suggest that VMP1 is important for the release of lipid droplets from the ER membrane to the cytosol because the number of ADRP-positive lipid droplets decreased in *VMP1*-silenced HepG2 cells (*Figure 6A,C*). Thus, VMP1 may regulate a common process shared by the three pathways derived from the ER: autophagy, lipoprotein formation, and lipid droplet formation. One hypothesis is that VMP1 might regulate remodeling of the ER membrane. The release of neural lipid-containing structures from the ER membrane to the ER lumen or the cytosol would require drastic reorganization of the membrane (*Figure 7D*). In addition, during autophagy, the ER membranes dramatically change their shapes and contact with the autophagic membranes (*Hayashi-Nishino et al., 2009*; *Zhao et al., 2017*; *Zhao et al., 2018*). Without VMP1, expansion of the autophagic membranes is defective (*Kishi-Itakura et al., 2014*; *Morita et al., 2018*). In *Drosophila* S2 cells (*Bard et al., 2006*) and in *Dictyostelium* (*Calvo-Garrido et al., 2008*), but not in *Caenorhabditis elegans* or in mammalian cells (*Zhao et al., 2017*), VMP1 was reported to be important for the secretion of some soluble cargos. Therefore, in some organisms, VMP1 may also be involved in the budding process of the ER membrane toward the Golgi, and this is achieved possibly by regulating the shape of the ER membrane. Since the release of lipoproteins and lipid droplets from the ER membrane can be regulated by lipid metabolism such as phospholipid remodeling (*Ben M'barek et al., 2017*; *Wang and Tontonoz, 2019*), VMP1 could play a possible role in lipid metabolism. Further investigations, in particular, structural analyses of VMP1 and its functionally related protein TMEM41B (*Moretti et al., 2018*; *Morita et al., 2018*; *Shoemaker et al., 2019*), as well as lipidomic analysis of cells lacking these factors, will reveal their molecular functions in the ER membrane.

In this study, we showed that VMP1 is essential for survival during larval periods and early embryonic periods in zebrafish and mice, respectively. *Vmp1*-deficient mice died around at 8.5 dpc. This timing of lethality is earlier than that of mice deficient for other core autophagy-related genes such as *Rb1cc1*, *Atg13*, and *Atg5* (*Kuma et al., 2017*; *Mizushima and Levine, 2010*). Considering the early embryonic lethality of *ApoB*- or *Mttp*-deficient mice around at 10.5 dpc due to malabsorption of lipids from maternal blood (*Farese et al., 1995*; *Farese et al., 1996*; *Raabe et al., 1998*), one of the causes of the early embryonic lethality of *Vmp1*-deficient mice is likely the same mechanism. The reduction of body weight (*Figure 3D*) and levels of serum cholesterol and lipoproteins (*Figure 3G*) in intestinal epithelial cell-specific *Vmp1*-deficient mice indicates defects in the absorption of nutrients. In contrast to intestinal epithelial cell-specific *Mttp*-deficient mice (*Iqbal et al., 2013*; *Xie et al., 2006*), intestinal epithelial cell-specific *Vmp1*-deficient mice showed milder phenotypes in lipoprotein secretion; the level of serum triglyceride did not decrease in intestinal epithelial cell-specific *Vmp1*-deficient mice (*Figure 3G*). Thus, although VMP1 is important, it is not absolutely essential for lipoprotein secretion.

Two independent genome-wide association studies in humans identified intronic single-nucleotide polymorphism associations (rs11650106 and rs2645492) in the *VMP1* gene with altered levels of circulating LDL (*Chu et al., 2012*; *Hoffmann et al., 2018*). Thus, VMP1 may be important for the regulation of the levels of circulating lipoproteins in humans. Further examination of the function of VMP1 in a whole organism would provide new insights into the regulation of lipid homeostasis under physiological as well as disease conditions.

# Materials and methods

## Key resources table

| Reagent type (species) or resource | Designation | Source or reference | Identifiers | Additional information |
|---|---|---|---|---|
| Genetic reagent (*D. rerio*) | *vmp1* | this paper | | |
| Genetic reagent (*D. rerio*) | *rb1cc1/fip200* | PMID: 27818143 | | |
| Genetic reagent (*D. rerio*) | *atg5* | this paper | | |
| Genetic reagent (*M. musculus*) | *Vmp1$^{-/-}$* | KOMP Repository | MGI allele Vmp1tm1a(KOMP)Wtsi, clone EPD0846_3_F07 | |
| Genetic reagent (*M. musculus*) | *Vmp1$^{flox/flox}$* | The European Mouse Mutant Archive | EMMA ID: EM05506 | |
| Genetic reagent (*M. musculus*) | Villin-*Cre* | Model Animal Research Center of Nanjing University | | |
| Cell line (*H. sapiens*) | HepG2 | ATCC | Cat. # HB-8065 RRID: CVCL_0027 | Negative for mycoplasma |
| Antibody | anti-ADRP (rabbit polyclonal) | Proteintech | Cat. #15294–1-AP | IF (1:200) |
| Antibody | anti-albumin (rabbit polyclonal) | Proteintech | Cat. #16475–1-AP, RRID: AB_2242567 | WB (1:1000) |
| Antibody | anti-APOA-I (mouse monoclonal) | Proteintech | Cat. #66206–1-Ig | WB (1:1000) |
| Antibody | anti-APOA-I (rabbit polyclonal) | Abcam | Cat. #ab64308 | IF (1:200) |
| Antibody | anti-APOB (goat polyclonal) | Rockland Immunochemicals Inc | Cat. #600-101-111, RRID: AB_2056958 | WB (1:1000) IF (1:200) |

*Continued on next page*

*Continued*

| Reagent type (species) or resource | Designation | Source or reference | Identifiers | Additional information |
|---|---|---|---|---|
| Antibody | anti-APOB (rabbit polyclonal) | Abcam | Cat. #ab20737, RRID: AB_2056954 | IHC (1:200) |
| Antibody | anti-APOE (mouse monoclonal) | Proteintech | Cat. #66830–1-Ig | WB (1:1000) IF (1:200) |
| Antibody | anti-α-tubulin (mouse monoclonal) | Sigma-Aldrich | Cat. #T9026, RRID: AB_477593 | WB (1:1000) |
| Antibody | anti-β-actin (mouse monoclonal) | Sigma-Aldrich | Cat. #A2228, RRID: AB_476697 | WB (1:1000) |
| Antibody | anti-BiP (rabbit polyclonal) | Abcam | Cat. #ab21685, RRID: AB_2119834 | WB (1:1000) |
| Antibody | anti-HERP (mouse monoclonal) | Chondrex | Cat. #7039 | WB (1:1000) |
| Antibody | anti-LC3 (mouse monoclonal) | Cosmo Bio | Cat. #CTB-LC3-2-IC | WB (1:1000) |
| Antibody | anti-LDH (rabbit monoclonal) | Abcam | Cat. #ab52488, RRID: AB_2134961 | WB (1:1000) |
| Antibody | anti-MTTP (mouse monoclonal) | Santa Cruz | Cat. #sc-135994, RRID: AB_2148288 | WB (1:1000) |
| Antibody | anti-p62 (rabbit polyclonal) | MBL International | Cat. #PM045, RRID: AB_1279301 | IHC (1:200) |
| Antibody | anti-PDI (mouse monoclonal) | Enzo Life Sciences | Cat. #ADI-SPA-891, RRID: AB_10615355 | WB (1:1000) |
| Antibody | anti-SEC61B (rabbit polyclonal) | Proteintech | Cat. #15087–1-AP, RRID: AB_2186411 | IHC (1:200) |
| Antibody | anti-VMP1 (rabbit polyclonal) | MBL International | Cat. #PM072 | WB (1:1000) |
| commercial assay or kit | Cholesterol Quantitation Kit | Biovision inc | Cat. #K603-100 | |
| commercial assay or kit | Cell-Based Proteasome-Glo Assays | Promega | Cat. #G8660 | |
| commercial assay or kit | Triglyceride Quantification Kit | Biovision inc | Cat. #K622-100 | |
| chemical compound, drug | BSA-conjugated oleic acid | Nacalai Tesque | Cat. #25630 | |
| chemical compound, drug | CP-346086 | Sigma-Aldrich | Cat. #PZ0103 | |
| chemical compound, drug | Lactacystin | Peptide Institute Inc | Cat. #4368-v | |
| chemical compound, drug | MG132 | Sigma-Aldrich | Cat. #M8699 | |
| chemical compound, drug | Thapsigargin | Sigma-Aldrich | Cat. #586005 | |
| chemical compound, drug | Tunicamycin | Sigma-Aldrich | Cat. #T7765 | |
| Other | BODIPY 558/568 $C_{12}$ | Thermo Fisher Scientific | Cat. #D3835 | |
| Other | 4',6-diamidino-2-phenylindole (DAPI) | Sigma-Aldrich | Cat. #D9542 | |
| Other | Hoechst33342 | Dojindo Molecular Technologies | Cat. #H342 | |

*Continued on next page*

*Continued*

| Reagent type (species) or resource | Designation | Source or reference | Identifiers | Additional information |
|---|---|---|---|---|
| Other | LipidTOX Red | Thermo Fisher Scientific | Cat. #H34476 | |
| Other | Nile red | Thermo Fisher Scientific | Cat. #N1142 | |
| Other | Oil red O | Sigma-Aldrich | Cat. #O0625 | |

## Zebrafish

RIKEN Wako wild-type strain was obtained from the Zebrafish National Bioresource Project of Japan, raised, and maintained in 14 hr light/10 hr dark conditions at 28.5°C according to established protocols (*Kimmel et al., 1995*). $Vmp1^{-/-}$ zebrafish were generated using the CRISPR/Cas9 system (*Jao et al., 2013*) including pT7-gRNA, a gift from Wenbiao Chen (plasmid #46759, Addgene), and Cas9 mRNA (CAS500A-1, System Biosciences). A region within exon 6 of zebrafish *vmp1* gene was targeted based on CRISPRscan (*Moreno-Mateos et al., 2015*) (target sequence was 5'-ccaTTGG TGAGCTGCCTCCATAC-3', where the protospacer adjacent motif is indicated by lower cases). gRNA was synthesized using a MEGAshortscript T7 transcription kit (AM1354, Thermo Fisher Scientific) and purified using a mirVana miRNA Isolation Kit (AM1560, Thermo Fisher Scientific). Wild-type embryos were microinjected at the one-cell stage with 100 pg of sgRNA and 300 pg of Cas9 mRNA using FemtoJet (Eppendorf) equipped with a Femtotip II injection capillary (Eppendorf). For genotyping of $vmp1^{-/-}$ zebrafish, heteroduplex mobility assay (*Ota et al., 2014*) was performed using genomic DNA, primers flanking the target site (forward primer, 5'-GCTCATCATTTGTACATGCG TGCGTG-3'; reverse primer, 5'-GCTCCAGCATCTCCTCGAATTCTTC-3'), PrimeSTAR Max DNA polymerase (R045A, TaKaRa Bio Inc), and 10% polyacrylamide gels. $Vmp1^{-/-}$ zebrafish were generated by intercrossing $vmp1^{+/-}$ zebrafish harboring a 7 bp deletion. $Rb1cc1^{-/-}$ and $atg5^{-/-}$ zebrafish were generated using $rb1cc1^{+/-}$ zebrafish harboring a 13 bp deletion in exon 4 and $atg5^{+/-}$ zebrafish harboring a 4 bp insertion in exon 3, respectively. Detailed descriptions of the phenotypes of the $rb1cc1^{-/-}$ and $atg5^{-/-}$ zebrafish will be reported elsewhere. Defects in autophagy in $rb1cc1^{-/-}$ zebrafish have been previously confirmed (*Kaizuka et al., 2016*).

A survival assay of zebrafish larvae was performed on progeny from intercrosses of $vmp1^{+/-}$ zebrafish in the same nursery environment without food. Dead larvae were collected twice a day and frozen. At 13 dpf, the remaining larvae were sacrificed, and all larvae including dead zebrafish were genotyped. Results are shown as Kaplan-Meier survival curves. The external appearance of zebrafish larvae was observed and imaged by a stereoscopic microscope (SZX10, Olympus).

## Mice

The $Vmp1^{gt/gt}$ mouse line was generated using the ES cell line Vmp1_F07 (CSD80081) containing an insertion of a gene trap (*gt*) cassette in the *Vmp1* gene (purchased from the Knockout Mouse Project Repository). ES cells were injected into C57BL/6 blastocysts to obtain chimeric mice, which were crossed with C57BL/6 mice to obtain heterozygous mutant mice. For genotyping of $Vmp1^{gt/gt}$ mice, genomic DNA was isolated from the tail or epiblasts dissected from the conceptus and amplified by PCR using primers (F1, 5'-CCCAAGTCTGCTTTACTGACAGCC-3'; F2, 5'-GGGATCTCATGCTGGAG TTCTTCG-3'; R, 5'-TTACTCAGACAGCCTTTCTCCACCC-3') to detect both 445 bp and 640 bp products for wild-type and *gt* alleles, respectively. The external appearance of mouse embryos was observed and imaged by a stereoscopic microscope (SZX10, Olympus). Wild-type C57BL/6 mice were obtained from Japan SLC, Inc.

The $Vmp1^{flox}$ mice were purchased from The European Mouse Mutant Archive (EM:05506). The exons 3 and 4 of *Vmp1* were flanked by two *loxP* sequences. Cre-mediated depletion of exons 3 and 4 leads to a frameshift, resulting in a small truncated peptide. The following primers were used to detect wild-type and floxed alleles of *Vmp1*: 5'-GCTTGCTGTGAATGGTTACC-3' (forward) and 5'-TCAGATCAGCCTTCTGTAGG-3' (reverse). The expected sizes are 266 bp and 391 bp, respectively. To generate intestinal epithelial cell-specific *Vmp1* knockout mice, $Vmp1^{flox/flox}$ mice were crossed

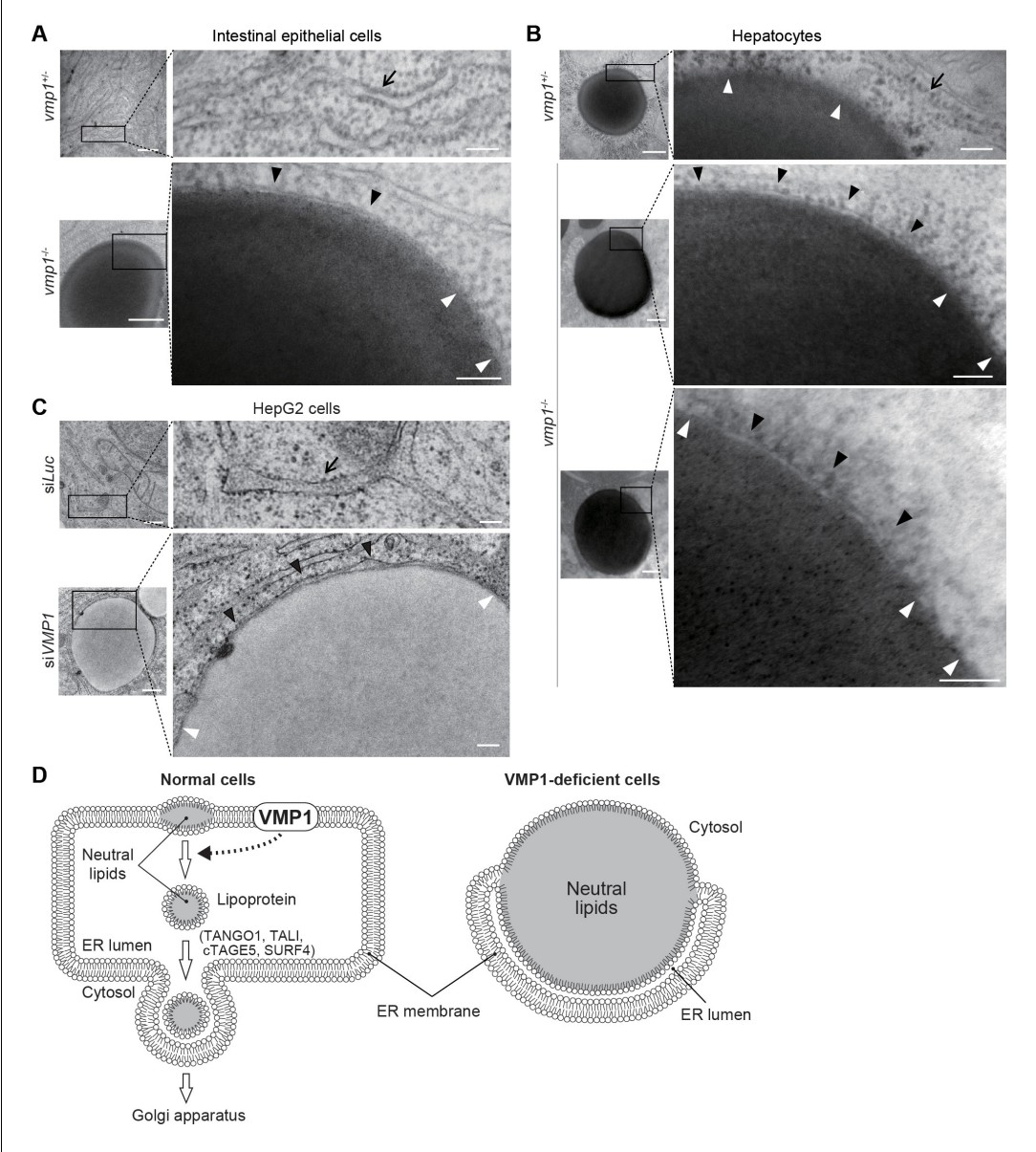

**Figure 7.** Neutral lipids accumulate within the ER membrane in the absence of VMP1. (A–C) Transmission electron microscopy of intestinal epithelial cells (A) and hepatocytes (B) from 6-dpf *vmp1*$^{+/-}$ and *vmp1*$^{-/-}$ zebrafish and VMP1-depleted HepG2 cells (C). Black and white arrowheads indicate the presence and absence of a lipid bilayer on neutral lipid-containing structures, respectively. Arrows indicate the ER membrane. Data are representative of three independent experiments. Scale bars, 500 nm and 100 nm in magnified panels. (D) Models for the membrane structure on lipids in the ER in wild-type and VMP1-deficient cells. Black and white arrowheads correspond to those in (A) to (C). In VMP1-deficient cells, the surfaces of neutral lipid structures (monolayer) are continuous to the ER membranes (bilayer), whereas only phospholipid monolayers cover neutral lipid structures in normal cells.

DOI: https://doi.org/10.7554/eLife.48834.021

with Villin-*Cre* mice (Model Animal Research Center of Nanjing University). Mice were maintained under specific pathogen-free conditions in the animal facility at the Institute of Biophysics, Chinese Academy of Sciences, Beijing.

All animal experiments were approved by the Institutional Animal Care and Use Committee of the University of Tokyo (Medical-P17-084) and the Institutional Committee of the Institute of Biophysics, Chinese Academy of Sciences (SYXK2016-35).

## Cell culture

HepG2 cells (HB-8065, ATCC) were cultured in Dulbecco's modified Eagle's medium (DMEM; D6546, Sigma-Aldrich) supplemented with 10% fetal bovine serum (172012, Sigma-Aldrich) and 2 mM L-glutamine (25030–081, Gibco; regular medium) in a 5% $CO_2$ incubator. HepG2 cells were regularly tested and found to be mycoplasma-free by DAPI DNA staining. For neutral lipid staining, cells were cultured with regular medium containing 10% serum and 1 µg/mL BODIPY 558/568 $C_{12}$ (D3835, Thermo Fisher Scientific) for 1 hr.

## RNA interference

Stealth RNAi oligonucleotide (Thermo Fisher Scientific) against human *VMP1* or *FITM2* was used for small interfering RNA (siRNA) experiments. The following sequences were used: human *VMP1* siRNA 5'-GCAUCAACAGUAUGUGCAACGUAUA-3' and human *FITM2* siRNA 5'- AAACAAGGUGCCAAA-CACCUUCUGG-3'. For the negative control, siRNA against *luciferase* (5'-CGCGGUCGGUAAAG UUGUUCCAUUU-3') (Thermo Fisher Scientific) was used. The Stealth RNAi oligonucleotides were transfected into cells using Lipofectamine RNAiMAX (13778–150, Thermo Fisher Scientific) according to the manufacturer's protocols. After 2 days, the cells were again transfected with the same siRNA and cultured for an additional 3 days before analysis.

## Antibodies and reagents

For immunoblotting, goat polyclonal anti-APOB (600-101-111, Rockland Immunochemicals Inc), rabbit polyclonal anti-VMP1 (PM072, MBL International), anti-Albumin (16475–1-AP, Proteintech), anti-BiP (ab21685, Abcam), mouse monoclonal anti-LC3 (CTB-LC3-2-IC, Cosmo Bio), anti-α-tubulin (T9026, clone DM1A, Sigma-Aldrich), anti-β-actin (A2228, clone AC-74, Sigma-Aldrich), anti-APOA-I (66206–1-Ig, clone 1C9G5, Proteintech), anti-APOE (66830–1-Ig, clone 1B2c9, Proteintech), anti-HERP (7039, clone HT2, Chondrex), anti-MTTP (sc-135994, Santa-Cruz), anti-PDI (ADI-SPA-891, clone 1D3, Enzo Life Sciences), and rabbit monoclonal anti-lactate dehydrogenase (LDH; ab52488, Abcam) antibodies were used as primary antibodies. Anti-goat (305-035-003), anti-mouse (115-035-003), and anti-rabbit (111-035-144) horseradish peroxidase-conjugated immunoglobulin G (IgG; Jackson ImmunoResearch Laboratories) were used as secondary antibodies. For immunostaining, goat polyclonal anti-APOB antibody and rabbit polyclonal anti-APOE, anti-ADRP/perilipin2 (15294–1-AP, Proteintech), and anti-APOA-I (ab64308, Abcam) antibodies were used as primary antibodies for the staining of culture cells. Rabbit polyclonal anti-APOB (ab20737, Abcam), anti-SEC61B (15087–1-AP, Proteintech), and anti-p62 (PM045, MBL International) antibodies were used for the staining of tissues. AlexaFluor 488-conjugated anti-goat IgG (A11055, Thermo Fisher Scientific), AlexaFluor 488-conjugated anti-rabbit IgG (A11008, Thermo Fisher Scientific), AlexaFluor 647-conjugated anti-rabbit IgG (A31573, Thermo Fisher Scientific), and AlexaFluor 647-conjugated anti-mouse IgG (A31571, Thermo Fisher Scientific) were used as secondary antibodies. For staining of the mouse intestine, FITC-conjugated anti-rabbit IgG (111-095-003, Jackson ImmunoResearch Laboratories) was used. Hoechst33342 (H342, Dojindo Molecular Technologies) and 4',6-diamidino-2-phenylindole (DAPI; D9542, Sigma-Aldrich) was used to stain DNA. LipidTOX Red (H34476, Thermo Fisher Scientific) was used to stain neutral lipids. Bovine serum albumin (BSA)-conjugated oleic acid (25630, Nacalai Tesque) was prepared as previously reported (*Velikkakath et al., 2012*). Lactacystin (4368 v) was purchased from the Peptide Institute Inc. MG132 (M8699), tunicamycin (T7765), thapsigargin (586005) and CP-346086 (PZ0103) were purchased from Sigma-Aldrich.

## Live imaging of zebrafish embryos

Zebrafish eggs at the one-cell stage were microinjected with 50 ng/µL of GFP-LC3 mRNA, which was synthesized from pcDNA3-GFP-LC3-RFP-LC3ΔG plasmid (*Kaizuka et al., 2016*) using a mMES-SAGE mMACHINE T7 Transcription Kit (AM1344, Thermo Fisher Scientific) and purified using RNeasy Mini Kit (74104, Qiagen). Embryos were anesthetized with 0.03% tricaine (A5040, Sigma-Aldrich), placed in water on a glass-bottomed dish, and viewed using a confocal microscope (FV1000 IX81; Olympus) with an objective lens (UPLSAPO30XS, Olympus).

## Immunohistochemistry

Immunohistochemistry of tissues was performed as described previously (*Morishita et al., 2013*). In brief, zebrafish larvae and mouse embryos dissected from the conceptus were fixed in 4% paraformaldehyde (PFA) overnight at 4℃, infiltrated with 15% and 30% sucrose in phosphate-buffered saline (PBS) for 4 hr each, and embedded in Tissue-Tek OCT Compound (Sakura Japan Co.). Sections (7 μm) were prepared using a cryostat (CM3050 S, Leica Microsystems) and mounted on slides. For whole-mount staining of mouse embryos, embryos were dissected from the conceptus and fixed with 4% PFA for 15 min at 4℃. Cryosections or mouse embryos were washed with PBS, treated with 0.05% Triton X-100 for 15 min, blocked with 3% BSA in PBS for 30 min, and incubated with primary antibodies for 1 hr, followed by PBS wash and incubation with secondary antibodies for 1 hr. For staining of neutral lipids and nuclear DNA, samples were treated with LipidTOX Red and Hoechst33342 in PBS for 30 min and washed three times with PBS. The coverslips and mouse embryos in a glass-bottomed dish were mounted with SlowFade antifade reagents (S36936, Thermo Fisher Scientific), viewed using a confocal laser microscope (FV1000 IX81, Olympus), and captured with FluoView software (Olympus). The number of punctate structures was determined using FIJI software (ImageJ, National Institutes of Health) (*Schindelin et al., 2012*).

For p62 and DAPI staining in intestinal epithelial cell-specific *Vmp1*-deficient mice, sections were deparaffinized in xylene and rehydrated in an ethanol series (100% × 3, 95%, and 75%). Antigen retrieval was performed using microwaves (0.01 M citrate buffer for 10 min). After blocking, sections were incubated with primary antibodies at 4℃ overnight. After washing three times in PBS, sections were incubated with fluorescent-labeled secondary antibodies for 1 hr at room temperature. Samples were then counterstained with DAPI and detected under a confocal microscope (LSM 880 Meta plus Zeiss Axiovert zoom, Zeiss).

## Oil red O and Nile red staining

Whole-mount oil red O staining was performed according to a previous method (*Dai et al., 2015*). In brief, zebrafish larvae were maintained in 0.2 mM 1-phenyl-2-thiourea (PTU) to avoid pigmentation from one dpf, fixed with 4% PFA overnight at 4℃, washed twice with PBS, infiltrated with 80% and 100% 1,2-propylene glycol for 30 min each, and stained with 0.5% oil red O (O0625, Sigma-Aldrich) in 100% 1,2-propylene glycol overnight at room temperature. Stained larvae were washed twice with PBS and the background color was faded with 100% and 80% 1,2-propylene glycol for 30 min each, and observed by a stereoscopic microscope (SZX10, Olympus). For oil red O staining of cryosections, zebrafish larvae were fixed in 4% PFA overnight at 4℃, infiltrated with 15% and 30% sucrose for 4 hr each at 4℃, and embedded in Tissue-Tek OCT Compound. Sections (6 μm) were mounted on slides and stained with oil red O in 60% isopropanol for 15 min at 37℃, followed by a 60% isopropanol wash, staining with hematoxylin for 3 min, and a wash with water. Slides were visualized using a microscope (BX51, Olympus) equipped with a digital camera (DP70, Olympus).

For Nile red staining in intestinal epithelial cell-specific *Vmp1*-deficient mice fed *ad libitum*, frozen tissues were embedded and cryostat sectioned. Sections were washed three times with PBS and then stained with Nile red (1:1000, N1142, Thermo Fisher Scientific) for 15 min at room temperature. Coverslips were mounted with DAPI and examined under a confocal microscope (LSM 880 Meta plus Zeiss Axiovert zoom, Zeiss).

## Alcian blue staining

Whole-mount Alcian blue staining to visualize the cartilage was performed according to the previous methods (*Walker and Kimmel, 2007*). In brief, zebrafish larvae were maintained in 0.2 mM PTU from one dpf, fixed with 4% PFA overnight at 4℃, dehydrated with 50% ethanol for 10 min, stained with acid-free stain solution (0.02% Alcian blue (A5268, Sigma-Aldrich), 60 mM MgCl$_2$, 70% ethanol) at room temperature overnight, cleared with 50% glycerol and 0.25% KOH at room temperature for 2 hr, dipped into 50% glycerol and 0.1% KOH, and viewed and photographed with a stereoscopic microscope (SZX10, Olympus).

## Immunocytochemistry

Cells grown on coverslips (S2441, Matsunami) were washed with PBS and fixed with 4% PFA for 15 min at room temperature. Fixed cells were permeabilized with 0.1% Triton X-100 (35501–15, Nacalai

Tesque) in PBS for 5 min and blocked with 3% BSA in PBS and incubated with specific antibodies for 1 hr. After washing with PBS, cells were incubated with Alexa Fluor 488- or 647-conjugated secondary antibodies for 1 hr. The coverslips were viewed using a confocal laser microscope (FV1000 IX81, Olympus) with a 100 × oil immersion objective lens (Olympus) and captured with FluoView software (Olympus). For the final output, the images were processed using Photoshop CS6 (Adobe). ImageJ software was used for quantification of the number and total pixel area of lipid-containing structures and the number of APOB- or ADRP-positive neutral lipid particles.

## Electron microscopy

Zebrafish larvae were dissected at the abdomen under anesthesia with 0.03% tricaine, and fixed with 2% glutaraldehyde and 2% PFA in 0.1 M sodium cacodylate buffer (0.1% calcium chloride) overnight. For HepG2 cells, cells were cultured on a poly-L-lysine coated cell tight C-2 cell disk (MS-0113K, Sumitomo Bakelite) and fixed in 2.5% glutaraldehyde (G015, TAAB) in 0.1 M phosphate buffer (pH 7.4) for 2 hr. Tissues and cells were then post-fixed with 1.0% osmium tetroxide in 0.1 M phosphate buffer for 2 hr, dehydrated, and embedded in Epon 812 according to a standard procedure. Ultrathin sections were stained with uranyl acetate and lead citrate and observed using an H-7100 electron microscope (Hitachi).

## Immunoblotting

Zebrafish embryos or HepG2 cells were lysed with lysis buffer (50 mM Tris-HCl [pH 7.5], 150 mM NaCl, 1 mM EDTA, 1% Triton X-100, 1 mM phenylmethanesulfonyl fluoride, and complete EDTA-free protease inhibitor cocktail [19543200, Roche]). After centrifugation at $15,000 \times g$ for 20 min, the supernatants were collected, and the protein concentrations were adjusted using the bicinchoninic acid method (23228, Thermo Fisher Scientific). The lysates were solubilized with immunoblot sample buffer (46.7 mM Tris-HCl [pH 6.8], 5% glycerol, 1.67% sodium dodecyl sulfate [SDS], 1.55% dithiothreitol, and 0.003% bromophenol blue). The immunoblot samples were separated by SDS-polyacrylamide gel electrophoresis, transferred to an Immobilon-P polyvinylidene difluoride membrane (IPVH00010, Millipore), and blotted with primary and secondary antibodies. Each protein signal was detected with Super-Signal West Pico Chemiluminescent substrate (1856136, Thermo Fisher Scientific) or Immobilon Western Chemiluminescent HRP substrate (WBKLS0500, Millipore). Signal intensities were captured using FUSION SOLO7S (Vilber-Lourmat). The images were processed using Photoshop CS6 (Adobe).

## Lipoprotein secretion assay

HepG2 cells were treated with siRNA twice as described above and cultured in serum-free DMEM supplemented with 200 nM oleic acids-BSA for 24 hr in 12-well plates. The cells were washed with PBS and re-cultured in serum-free DMEM before analysis. The culture medium was centrifuged at $5000 \times g$ for 3 min to remove any cells or cellular debris. The culture medium was then precipitated with 10% trichloroacetic acid (TCA), and the pellets were washed with ice-cold acetone twice and dissolved in immunoblot sample buffer. The cells were washed with PBS and lysed as described above. Samples (approximately 7% or 14% vol of total precipitated media or cell lysates, respectively) were subjected to immunoblot analysis. ImageJ software was used for densitometric quantification.

For chromatography analysis, siRNA-treated HepG2 cells (approximately $1.5 \times 10^6$ cells) were cultured in 0.1% BSA containing DMEM with or without oleic acids-BSA for 48 hr before analysis. Lipoproteins in culture medium were analyzed using the gel permeation high performance liquid chromatography (Skylight Biotech Inc) system as previously described (*Okazaki and Yamashita, 2016*). Briefly, lipoproteins in culture medium were separated with tandemly connected Skylight PakLP1-AA gel permeation columns (Skylight Biotech Inc; 300 mm × 4.6 mm I.D.). The column effluent was then equally split into two lines by a micro splitter, and each effluent was allowed to react at 37°C with the cholesterol and triglyceride reagents. Absorbance at 550 nm was continuously monitored after each enzymatic reaction in two reactor coils (PTFE; 25 m × 0.18 mm I.D.).

## Triglyceride and cholesterol quantification

Six pairs of 18-month-old $Vmp1^{flox/+}$;Villin-Cre mice and $Vmp1^{flox/flox}$;Villin-Cre mice fed *ad libitum* were examined. Blood was allowed to stand for 2 hr and centrifuged at 1600 *g* for 15 min for collecting the serum. Serum samples were then measured by OLYMPUS AU480 automatic biochemical analyzer.

HepG2 cells were treated with siRNA twice as described above. HepG2 cells (approximately $1 \times 10^5$ cells) were cultured in serum-free medium for 24 hr before analysis. For total lipid extraction from culture medium, the Bligh and Dyer method was performed. Both extra- and intra-cellular cholesterol and triglyceride levels were measured using quantitation kits (K603-100 and K622-100, respectively, Biovision Inc) according to the manufacturer's protocols.

## Measurement of proteasome activity

Proteasomal activity was measured with the Proteasome-Glo kit (G8660, Promega). HepG2 cells were treated with siRNA twice as described above and plated equally (approximately $1 \times 10^4$ cells per well) in 96-well white-walled plates. The cells were cultured with or without 5 µM lactacystin for 2 hr before measurement. Proteasome-Glo Cell-Based Reagent was prepared as per manufacturer's protocol and an equal volume was added to each well. The plate was mixed for 2 min using TAITEC E-36 micromixer and then incubated at room temperature for 10 min. Luminescence was measured by a microplate leader EnSpire (2300–00J, PerkinElmer).

## Quantitative real-time PCR

Total RNA was extracted from HepG2 cells using ISOGEN (319–90211, Nippon Gene) and reverse-transcribed using ReverTraAce (FSQ-201, TOYOBO) according to the manufacturer's instructions. PCR was performed by a Thermal Cycler Dice TP800 (TaKaRa Bio Inc) in triplicate using TB Green Premix Ex Taq II (RR820, TaKaRa Bio Inc). The expression level of *FITM2* was normalized to that of *ACTB*. Primers used are listed as follows: *FITM2*, 5′-AAAGGAACACCAGAGCAAGC-3′ and 5′-CCTCATGCAGCACAGACATC-3′; *ACTB*, 5′-ATTGCCGACAGGATGCAGAA-3′ and 5′-ACATCTGCTGGAAGGTGGACAG-3′.

## Statistical analysis

Student *t*-test or Mann-Whitney U-test were performed to compare two groups. All data are presented as the mean ± standard error of the mean unless otherwise stated. Statistical analysis was performed using R software (R Core Team) or GraphPad Prism seven software (GraphPad software).

## Acknowledgements

We thank Hong Zhang for continuous encouragement and support, Kota Saito for help with the secretion assays, Nozomi Sato and Tomoya Eguchi for care of the zebrafish and mice, Akira Ohtsuka and Akiko Kuma for help in establishing the mouse genotyping protocols, Keiko Igarashi for help with the histological assays, Chieko Saito for help with electron microscopy, and Yuki Ohsaki, Toyoshi Fujimoto, Takeshi Sugawara, and Hayashi Yamamoto for the helpful discussions.

## Additional information

### Competing interests

Noboru Mizushima: Reviewing editor, *eLife*. Mitsuyo Okazaki: received a consultant fee from Skylight Biotech Inc. but is not an employee of the company. The patent by M Okazaki (WO/2015/152371) belongs to Skylight Biotech Inc. The other authors declare that no competing interests exist.

### Funding

| Funder | Grant reference number | Author |
| --- | --- | --- |
| Japan Science and Technology Agency | JPMJER1702 | Noboru Mizushima |

| Japan Society for the Promotion of Science | 25111005 | Noboru Mizushima |
|---|---|---|
| National Natural Science Foundation of China | 31671430 | Yan G. Zhao |
| Japan Society for the Promotion of Science | 18K14694 | Hideaki Morishita |

The funders had no role in study design, data collection and interpretation, or the decision to submit the work for publication.

## Author contributions

Hideaki Morishita, Conceptualization, Data curation, Formal analysis, Funding acquisition, Validation, Investigation, Visualization, Methodology, Writing—original draft, Project administration, Writing—review and editing, Carried out most of the experiments using zebrafish and mouse embryos; Yan G Zhao, Conceptualization, Data curation, Formal analysis, Funding acquisition, Validation, Investigation, Visualization, Methodology, Writing—original draft, Project administration, Writing—review and editing, Carried out experiments using intestinal epithelial cell-specific Vmp1-deficient mice; Norito Tamura, Conceptualization, Data curation, Formal analysis, Validation, Investigation, Visualization, Methodology, Writing—original draft, Project administration, Writing—review and editing, Carried out most of the experiments using HepG2 cells, Assisted the experiments using mouse embryos; Taki Nishimura, Conceptualization, Data curation, Formal analysis, Validation, Investigation, Visualization, Methodology, Project administration, Writing—review and editing, Carried out the experiments using HepG2 cells; Yuki Kanda, Data curation, Validation, Investigation, Visualization, Methodology, Project administration, Writing—review and editing, Carried out the experiments using zebrafish; Yuriko Sakamaki, Data curation, Formal analysis, Validation, Investigation, Visualization, Methodology, Writing—review and editing, Assisted with electron microscopy studies; Mitsuyo Okazaki, Data curation, Formal analysis, Validation, Investigation, Visualization, Methodology, Writing—review and editing, Carried out HPLC analysis of lipoproteins in culture medium; Dongfang Li, Conceptualization, Data curation, Formal analysis, Validation, Investigation, Visualization, Methodology, Project administration, Writing—review and editing, Carried out experiments using intestinal epithelial cell-specific Vmp1-deficient mice; Noboru Mizushima, Conceptualization, Data curation, Formal analysis, Supervision, Funding acquisition, Validation, Investigation, Visualization, Methodology, Writing—original draft, Project administration, Writing—review and editing

## Author ORCIDs

Hideaki Morishita https://orcid.org/0000-0003-0860-8371
Yan G Zhao https://orcid.org/0000-0002-4588-6752
Norito Tamura https://orcid.org/0000-0001-5945-3117
Taki Nishimura https://orcid.org/0000-0003-4019-5984
Yuki Kanda https://orcid.org/0000-0003-0240-604X
Noboru Mizushima https://orcid.org/0000-0002-6258-6444

## Ethics

Animal experimentation: All animal experiments were approved by the Institutional Animal Care and Use Committee of the University of Tokyo (Medical-P17-084) and the Institutional Committee of the Institute of Biophysics, Chinese Academy of Sciences (SYXK2016-35).

## Decision letter and Author response

Decision letter https://doi.org/10.7554/eLife.48834.024
Author response https://doi.org/10.7554/eLife.48834.025

## Additional files

### Supplementary files

• Transparent reporting form
DOI: https://doi.org/10.7554/eLife.48834.022

### Data availability

All data generated or analysed during this study are included in the manuscript files. Source data files have been provided for Figures (1, 3, 4, 5, and 6), Figure 4—figure supplement 1, Figure 6—figure supplement 1, and Figure 6—figure supplement 2.

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
