## [Decision Letter]

[Editors’ note: a previous version of this study was rejected after peer review, but the authors submitted for reconsideration. The first decision letter after peer review is shown below.]

Thank you for submitting your work entitled "VMP1 is essential for release of lipoproteins from the endoplasmic reticulum membrane" for consideration by *eLife*. Your article has been reviewed by three peer reviewers, including Suzanne Pfeffer as the Reviewing Editor and Reviewer #1, and the evaluation has been overseen by a Senior Editor. The following individual involved in review of your submission has agreed to reveal their identity: M. Mahmood Hussain (Reviewer #3).

Our decision has been reached after consultation between the reviewers. Based on these discussions and the individual reviews below, we regret to inform you that your work will not be considered further for publication in *eLife*.

This study addresses the role of the ER protein VMP1 in lipoprotein biogenesis. The first half of demonstrates that VMP1 deficiency causes embryonic lethality and the accumulation of neutral lipids in intestinal epithelial cells and hepatocytes of zebrafish and mice. The reviewers all agreed that this part of the study is well done and convincing. However, while the idea that VMP1 mediates lipoprotein release from the ER is consistent with the results, more work would be necessary to make a convincing case, likely requiring more than a few months.

One reviewer noted that the Zhang group has published compelling evidence that VMP1 regulates the ER calcium channel ATP2A/SERCA (Zhao and Zhang, Autophagy, 2018; Zhao et al., 2017) and disrupting ER calcium homoeostasis might indirectly affect lipoprotein biogenesis. In addition, the aberrant lipid droplet structures found in HepG2 cells depleted of VMP1 have been seen when the cells are treated with proteasome inhibitors, which indicates that they do not only form when lipoprotein biogenesis is directly impacted. Therefore, the reviewers felt it would be important to add data to show that the role is not indirect. This could be addressed by more broadly assessing ER function, ER stress, and lipid metabolism after VMP1 knockdown. It would also be good to know whether inducing ER stress by other mechanism causes similar lipid droplet structures to form in cells or whether the abnormal LD formation is specific to depletion of VMP1 or other proteins involved in lipoprotein biogenesis like MTTP. Another good way to tell whether VMP1 depletion specifically affects lipoprotein biogenesis would be to measure the initial rate of APOB secretion and measure secretion rates for control proteins like APOA-I or albumin. A paper addressing these issues could surely be re-evaluated as a new submission in the future, but we must decline the manuscript at this stage since significant additional work would be required.

*Reviewer #1:*

This is a clear and well written paper that reports that VMP1 participates in lipoprotein biogenesis in the ER of zebrafish and mouse liver, intestine and visceral endoderm. In HepG2 cells, the authors see decreased secretion of triglycerides and cholesterol and a decrease in the lipoprotein marker, APOB. One experiment will add support to the conclusion that VMP1 acts early in the ER to drive release of nascent lipoprotein from the ER membrane and subsequent secretion: the authors should test the relative initial rate of APOB secretion from HepG2 cells ± VMP1. A 24 hour collection is more complicated by the altered regulation of APOBprotein seen in these cells (more rapid turnover) but the absolute rate of secretion can be monitored by 1, 2 and 4 hour time points for example and will provide a better metric of how important this protein is for lipoprotein secretion itself. This straightforward experiment will add much to the significance of this interesting story that should otherwise be published without delay in *eLife*.

Increased LC3-II is referred to as "defective autophagy" but perhaps it reflects increased autophagic flux?

*Reviewer #2:*

This study addresses the role of the ER protein VMP1 in lipoprotein biogenesis. The first half of demonstrates that VMP1 deficiency causes embryonic lethality and the accumulation of neutral lipids in intestinal epithelial cells and hepatocytes of zebrafish and mice. The part of the study is well done and convincing. The second half argues that VMP1 facilitates release of lipoproteins from the ER membrane into the lumen of the ER. However, the study does not make a convincing case that VMP1 directly affects lipoprotein biogenesis or even that there is a defect in lipoprotein release into the ER lumen. Since very little is known about this process, it is not clear whether the APOB-/ADRP-positive lipid droplet structures are stalled intermediates in lipoprotein biogenesis or are off-pathway products, caused by factors such as changes in lipid metabolism or defects in lipid droplet emergence into the cytoplasm. In Ohsaki et al. (2008), the Fujimoto group showed that adding proteosome inhibitors to Huh7 cells (and they say HepG2 cells) induces formation of similar structures that are positive for both APOB and ADRP. Indeed, they see these structures even in about 10% of untreated cells. It seems possible that removing VMP1 from cells could cause ER stress or change lipid metabolism and similarly indirectly cause the formation of APOB-/ADRP-positive structures.

*Reviewer #3:*

The investigators have generated VMP1 deficient zebrafish and mice and observed that this deficiency results in embryonic lethality. Using confocal microscopy they showed that APOB accumulates in cells. The major conclusion of the paper is that VMP1 is involved in desorption of LD from ER membrane. This conclusion is based on EM pictures presented in Figure 7. Investigators observed bulging of ER membrane in siVMP1 treated HepG2 cells. This suggests that some LD/lipoproteins assimilate in the ER membrane. Hence, they conclude that VMP1 helps in the dissociation of lipoproteins from the ER membrane. This is a novel finding of the paper and a novel function for *Vmp1*.

Plasma lipid and lipoprotein profile and lipid absorption studies in intestine-specific VMP1 KO mice would be required to ascertain the physiological importance of this protein in lipid absorption and lipoprotein assembly. Embryonic lethality can be attributed to VMP1's role in autophagy.

The paper provides convincing evidence that deficiency of VMP1 reduces APOB secretion in HepG2 cells. However, controls for the specificity are needed, such as secretion of APOA-I or albumin.

How does VMP1 help in the release of lipoproteins from the ER membrane? Does VMP1 interact with APOB? Does it interact with MTP? What happens to MTP in the absence of VMP1?

[Editors’ note: what now follows is the decision letter after the authors submitted for further consideration. The authors were asked to provide a plan for revisions before the editors issued a final decision. The authors’ plan for revisions was approved and the authors made a formal revised submission.]

Thank you for sending your article entitled "VMP1 is essential for release of lipoproteins from the endoplasmic reticulum membrane" for peer review at *eLife*. Your article is being evaluated by three peer reviewers, including Suzanne Pfeffer as the Reviewing Editor, and the evaluation is being overseen by Vivek Malhotra as the Senior Editor.

Given the list of comments, the editors and reviewers invite you to respond with an action plan and timetable for the completion of the additional work. We plan to share your responses with the reviewers and then issue a binding recommendation.

Reviewer 1's concerns are the most important:

"I am still not convinced that VMP1 knockdown reduces the rate or extent of VLDL secretion from HepG2 cells. The evidence is inconsistent. Figure 4C, D suggests that there is a profound defect but is hard to reconcile with the data in Figure 4A, B and E, which suggest little change. In Figure 4E, the amount of APO proteins secreted after VMP knockdown is normalized to controls but it seems more appropriate to normalize it to the amount of APO remaining in the cells. When expressed in this way, there does not seem to be any decrease in the relative amount or rate of secretion of any of the APO proteins. How can this be reconciled with the evidence in Figure 4C, D?"

Reviewer 3 felt that the conclusions went beyond the actual data and need to be toned down and clarified (see below); they wrote, "The data indicate that VMP1 is not "essential" for APOB-lipoprotein assembly. If it was essential then we would have seen a more robust phenotype. It may play a role in lipoprotein secretion. But it is not a specific role. It probably plays a general role in secretion. This is based on the observation that there is reduction in APOB, APOA-I and APOE. This is not a new finding. The authors should be realistic in interpretation of their data."

*Reviewer #1:*

The authors have tried to respond with care to each of the reviewer comments. In looking at the rate of release of APOB and APOE, they find APOB levels decrease but a clear decrease for APOE. The legend does not explain how the gels were loaded and needs to be clarified. Also, it would be clearer for the reader if the authors graphed percent of total apolipoprotein in medium on a graph with time as a linear function.

*Reviewer #3:*

– To prove definitely that VMP1 is essential for release of lipoproteins from the ER membranes, authors must follow APOB along with lipids. They should demonstrate that in the absence of VMP1, APOB remains in the ER membrane and is not present in the ER lumen. Thus, distribution of APOB and other control proteins in ER lumen and ER membrane must be studied. [Is there a way for you to account for APOB in support of your conclusions?]

– In absence of APOB-lipoprotein assembly, secretion of APOA-I is largely unaffected. Thus, significant decrease in APOA-I secretion in these studies suggests that VMP1 may have a generalized effect on secretion of soluble proteins as cited in the third paragraph of the Introduction. Thus, the argument that VMP1 is specific for APOB-lipoproteins is weak. APOA-I is secreted independent of APOB-lipoproteins. Its secretion does not require removal from ER membrane.

– Figure 5A: Is Sec61 surrounding the lipid droplet? It looks like LDs are in the lumen of ER. This suggests that they have been released from the ER membrane.

– EM is required to show that VMP1 deficiency leads to the accumulation of LDs and APOB (immune-gold) in the ER membrane if the hypothesis is that VMP1 is critical for the release of lipoprotein from the ER membrane. [Without such EM, can the conclusions be softened?]

– Need separation of ER lumen and membrane and distribution of lipids and APOB in these fractions [if you want to conclude that the protein actually releases lipoprotein from ER membrane].

– It is likely that primordial lipoproteins are formed and they are in the ER lumen. VMP1 deficiency may interfere with their further transport to Golgi.

– ADRP^+^ and APOB^+^ lipid droplets may be cytosolic droplets that contain APOB and may be unique to VMP1 deficiency.

– This paper suggest that VMP1 is essential for the release of lipoproteins from the ER membrane, but the intestine-specific ablation phenotype on plasma is under whelming. Therefore, it is possible that VMP1 may play a role in lipoprotein secretion but it is not essential. Further, VMP1 deficiency appears to affect all apolipoprotein secretion. The authors are trying to over interpret their results. They should compare the phenotype in mice that are deficient in APOB and MTP; the two proteins known to be essential for B-lipoprotein assembly and secretion.

– It is unclear why LDL and HDL are decreased. How did they measure LDL and HDL? Why cholesterol decreased but no change in triglyceride? This phenotype does not support the idea that VMP1 is essential for APOB-lipoprotein assembly and secretion. Most likely VMP1 is involved in general secretory pathway. Its deficiency slows down secretion of APOB. Similar thing is happening for APOA-I and APOE. Thus, there is little specificity to this process.

– VMP1 also affects SERCA. Can the phenotype be explained by alterations in calcium pump in the ER?

– Was blood collected from fasted mice?

– Figure 3: Why there is a significant decrease in HDL? If VMP1 plays a role in APOB-lipoprotein assembly then it should not affect HDL levels. These data demand fat absorption studies in these mice to see if intestinal deficiency of VMP1 really has any significant defect in lipid absorption.

---

## [Author Response]

[Editors’ note: the author responses to the first round of peer review follow.]

This study addresses the role of the ER protein VMP1 in lipoprotein biogenesis. The first half of demonstrates that VMP1 deficiency causes embryonic lethality and the accumulation of neutral lipids in intestinal epithelial cells and hepatocytes of zebrafish and mice. The reviewers all agreed that this part of the study is well done and convincing. However, while the idea that VMP1 mediates lipoprotein release from the ER is consistent with the results, more work would be necessary to make a convincing case, likely requiring more than a few months.One reviewer noted that the Zhang group has published compelling evidence that VMP1 regulates the ER calcium channel ATP2A/SERCA (Zhao and Zhang, Autophagy, 2018; Zhao et al., 2017) and disrupting ER calcium homoeostasis might indirectly affect lipoprotein biogenesis. In addition, the aberrant lipid droplet structures found in HepG2 cells depleted of VMP1 have been seen when the cells are treated with proteasome inhibitors, which indicates that they do not only form when lipoprotein biogenesis is directly impacted. Therefore, the reviewers felt it would be important to add data to show that the role is not indirect. This could be addressed by more broadly assessing ER function, ER stress, and lipid metabolism after VMP1 knockdown.

We found no changes in the proteasome activity, ER stress, and the MTTP protein level in VMP1-depleted HepG2 cells (Figure 6—figure supplement 1A-C). These results suggest that the function of the ER is not affected broadly.

It would also be good to know whether inducing ER stress by other mechanism causes similar lipid droplet structures to form in cells or whether the abnormal LD formation is specific to depletion of VMP1 or other proteins involved in lipoprotein biogenesis like MTTP.

Crescent-shaped accumulations of APOB and ADRP around neutral lipids were not observed in HepG2 cells by treatment with proteasome inhibitors, ER stress inducers, or an MTTP inhibitor (Figure 6—figure supplement 2A-C) or by depletion of FITM2, a factor required for budding of lipid droplets from the ER to the cytosol (Figure 6—figure supplement 2D, E). Thus, as far as we have tested, the observed lipoprotein phenotype is rather specific to depletion of VMP1.

Another good way to tell whether VMP1 depletion specifically affects lipoprotein biogenesis would be to measure the initial rate of APOB secretion and measure secretion rates for control proteins like APOA-I or albumin.

We now show defects in secretion of APOB, APOE, and APOA-I, but not albumin, in VMP1-depleted HepG2 cells at early time points (Figure 4E).

Other major experimental data that we have added are:

- The levels of serum cholesterols and lipoproteins such as HDL decreased in intestinal epithelial cell-specific *Vmp1* KO mice (Figure 3G).

- Lipid lens structures were not detected in the ER membrane in wild-type zebrafish and HepG2 cells (Figure 7A-C).

Reviewer #1:This is a clear and well written paper that reports that VMP1 participates in lipoprotein biogenesis in the ER of zebrafish and mouse liver, intestine and visceral endoderm. In HepG2 cells, the authors see decreased secretion of triglycerides and cholesterol and a decrease in the lipoprotein marker, APOB. One experiment will add support to the conclusion that VMP1 acts early in the ER to drive release of nascent lipoprotein from the ER membrane and subsequent secretion: the authors should test the relative initial rate of APOB secretion from HepG2 cells ± VMP1. A 24 hour collection is more complicated by the altered regulation of APOB protein seen in these cells (more rapid turnover) but the absolute rate of secretion can be monitored by 1, 2 and 4 hour time points for example and will provide a better metric of how important this protein is for lipoprotein secretion itself. This straightforward experiment will add much to the significance of this interesting story that should otherwise be published without delay in eLife.

We would like to thank the reviewer for this suggestion. As recommended, we have performed immunoblotting of APOB at 0, 1, 2, 4, and 24 h time points following 24-h oleic acid treatment, and found that the secretion of APOB was impaired in VMP1-depleted HepG2 cells from early time points. We have also found that secretion of APOE and APOA-I, which associate with lipoproteins in the ER in HepG2 cells (Chisholm et al., 2002, J Lipid Res; Fazio and Yao, 1995, Arterioscler Thromb Vasc Biol; Sundaram and Yao, 2010), was significantly impaired at early time points in VMP1-depleted HepG2 cells. Secretion of albumin, which is transported from the ER to Golgi separately from lipoproteins (Tiwari and Siddiqui, 2012), was slightly but not significantly impaired in VMP1-silenced HepG2 cells especially at early time points when secretion of apolipoproteins was significantly suppressed. Thus, a defect in secretion in VMP1-deficient cells is not general, but rather specific to lipoproteins. These results and discussion have been included in new Figure 4E and the text in the revised manuscript.

Increased LC3-II is referred to as "defective autophagy" but perhaps it reflects increased autophagic flux?

If autophagic flux is blocked at a step downstream of LC3 lipidation, LC3-II should accumulate because of reduced turnover. This is the case in VMP1-depleted cells. In fact, recent studies using Vmp1-deficient HEK293T, HeLa, or MEF cells demonstrated that autophagic flux is completely suppressed in Vmp1-deficient cells that accumulate LC3-II (Itakura and Mizushima, 2010; Morita et al. 2018; Shoemaker et al., 2019; Zhao et al., 2017). We have included these references in the revised manuscript.

Reviewer #2:This study addresses the role of the ER protein VMP1 in lipoprotein biogenesis. The first half of demonstrates that VMP1 deficiency causes embryonic lethality and the accumulation of neutral lipids in intestinal epithelial cells and hepatocytes of zebrafish and mice. The part of the study is well done and convincing. The second half argues that VMP1 facilitates release of lipoproteins from the ER membrane into the lumen of the ER. However, the study does not make a convincing case that VMP1 directly affects lipoprotein biogenesis or even that there is a defect in lipoprotein release into the ER lumen. Since very little is known about this process, it is not clear whether the APOB-/ADRP-positive lipid droplet structures are stalled intermediates in lipoprotein biogenesis or are off-pathway products, caused by factors such as changes in lipid metabolism or defects in lipid droplet emergence into the cytoplasm. In Ohsaki et al. (2008), the Fujimoto group showed that adding proteosome inhibitors to Huh7 cells (and they say HepG2 cells) induces formation of similar structures that are positive for both APOB and ADRP. Indeed, they see these structures even in about 10% of untreated cells. It seems possible that removing VMP1 from cells could cause ER stress or change lipid metabolism and similarly indirectly cause the formation of APOB-/ADRP-positive structures.

We would like to thank the reviewer for valuable suggestions. As this reviewer mentions, very little is known about the process of lipoprotein biogenesis, particularly at the step of releasing into the ER lumen. We understand that it would be ideal to prove that VMP1 directly regulates this process, but this is difficult using currently available information. Given this situation, we believe that providing the first evidence that the ER protein VMP1 is required for the step is important and informative for the field. Nevertheless, we have tried to rule out secondary possibilities as much as possible as following.

To exclude the possibility that crescent-shaped accumulations of APOB and ADRP are secondarily induced by proteasome inhibition, we have evaluated the activity of proteasome and found that proteasome activity was not suppressed in VMP1-depleted HepG2 cells. Also, as opposed to the previous report by Fujimoto’s group (Ohsaki et al., 2008), we hardly observed the crescent-shaped accumulations of APOB and ADRP in wildtype HepG2 cells irrespective of treatment with proteasome inhibitors (MG132 and lactacystin) (0% of randomly selected cells (n ≥ 39) with or without proteasome inhibitors). Furthermore, VMP1-depleted HepG2 cells did not demonstrate ER stress and reduced MTTP protein level, which can induce APOB proteolysis, and that treatment of wild-type HepG2 cells with ER stress inducers (tunicamycin and thapsigargin) or an MTTP inhibitor (CP346086) did not induce the crescent-shaped accumulations of APOB and ADRP. These results suggest that the crescent-shaped accumulations of APOB and ADRP is independent of proteasome inhibition, ER stress, and MTTP inhibition in HepG2 cells. These data have been included in the revised manuscript in new Figure 6—figure supplement 1A-C and Figure 6—figure supplement 2A-C.

To determine whether the crescent-shaped accumulations of APOB and ADRP are induced as a result of a defect in lipid droplet emergence into the cytosol, we have depleted FITM2 that is required for budding of lipid droplets (Kadereit et al., 2008; Choudhary et al., 2015) but not for lipoprotein secretion (Goh et al., 2015). Deletion of FITM2 in HepG2 cells caused a reduction of the total volume of lipid droplets as reported in FITM2-depleted adipocytes, zebrafish, and *C. elegans* (Kadereit et al., 2008; Choudhary et al., 2015). However, the crescent-shaped accumulations of APOB and ADRP were not observed in FITM2-depleted HepG2 cells. Thus, it is unlikely that defective budding of lipid droplets from the ER membrane causes the impairment of lipoprotein budding in VMP1-depleted cells. These data have been included in the revised manuscript in new Figure 6—figure supplement 2D, E.

We agree with the comment that crescent-shaped accumulations of APOB and ADRP may be induced as a result of an alteration of lipid metabolism. Indeed, recent studies have suggested that release of lipoproteins and lipid droplets from the ER membrane is regulated by lipid metabolism such as phospholipid remodeling (Ben M'barek et al., 2017; Wang and Tontonoz, 2019), suggesting the possible role of VMP1 in lipid metabolism. More extensive investigation, including lipidomic analyses of VMP1-depleted cells and mechanistic in vitro assays, would be required to examine this hypothesis, which we believe would be a separate complete work. We have discussed this point in the second paragraph of Discussion.

Reviewer #3:The investigators have generated VMP1 deficient zebrafish and mice and observed that this deficiency results in embryonic lethality. Using confocal microscopy they showed that APOB accumulates in cells. The major conclusion of the paper is that VMP1 is involved in desorption of LD from ER membrane. This conclusion is based on EM pictures presented in Figure 7. Investigators observed bulging of ER membrane in siVMP1 treated HepG2 cells. This suggests that some LD/lipoproteins assimilate in the ER membrane. Hence, they conclude that VMP1 helps in the dissociation of lipoproteins from the ER membrane. This is a novel finding of the paper and a novel function for Vmp1.Plasma lipid and lipoprotein profile and lipid absorption studies in intestine-specific VMP1 KO mice would be required to ascertain the physiological importance of this protein in lipid absorption and lipoprotein assembly.

We would like to thank the reviewer for valuable suggestions. As recommended, we quantified the amount of serum triglyceride, cholesterol, LDL, and HDL in intestinal epithelial cell-specific *Vmp1* KO mice, and found that levels of cholesterol and HDL were significantly reduced in *Vmp1* KO mice. In contrast, the level of serum triglyceride did not decrease. We speculate that this could be due to a possible compensatory mechanism. We have included the data in new Figure 3G and discussed this point in the last paragraph of Discussion.

Embryonic lethality can be attributed to VMP1's role in autophagy.

As we have already discussed in the original version, the timing of lethality of *Vmp1* KO mice (around 8.5 dpc) is earlier than that of mice deficient for other core autophagy-related genes such as *Rb1cc1* (around 16 dpc)*, Atg13* (around 17 dpc), and *Atg5* (around 0.5 day after birth) (Kuma et al., 2017; Mizushima and Levine, 2010). Thus, we think that the early embryonic lethality of *Vmp1* KO mice is due to defects in autophagy-independent pathways.

The paper provides convincing evidence that deficiency of VMP1 reduces APOB secretion in HepG2 cells. However, controls for the specificity are needed, such as secretion of APOA-I or albumin.

We thank the reviewer for pointing this out. As recommended, we have performed immunoblotting of APOB, APOE, APOA-I, and albumin at 0, 1, 2, 4, and 24 h time points following 24-h oleic acid treatment, and found that the secretion of not only APOB but also APOE and APOA-I was significantly impaired at early time points in VMP1-depleted HepG2 cells. Secretion of albumin, which is transported from the ER to Golgi separately from lipoproteins (Tiwari and Siddiqui, 2012), was slightly but not significantly impaired in VMP1-silenced HepG2 cells especially at early time points. Thus, a defect in secretion in VMP1-deficient cells is not general, but rather specific to lipoproteins. These results and discussion have been included in new Figure 4E and the text in the revised manuscript.

How does VMP1 help in the release of lipoproteins from the ER membrane? Does VMP1 interact with APOB? Does it interact with MTP? What happens to MTP in the absence of VMP1?

As suggested by the reviewer, we have performed immunoprecipitation assays and found that FLAG-VMP1 interacted with endogenous APOB, but not MTTP, in HepG2 cells (Author response image 1). However, this interaction was weak and only detected when we used the detergent Triton X-100 (1%), but not dodecyl maltoside (1%)/cholesteryl hemisuccinate (0.2%). Thus, more extensive investigation would be required to prove this potential interaction and its physiological relevance, which we believe would be a separate complete work.

To investigate the effect of VMP1 deficiency on MTTP, we have performed immunoblotting of MTTP and found that there was no difference in the protein level of MTTP between wild-type and VMP1-depleted HepG2 cells. Also, crescent-shaped accumulations of APOB and ADRP were not observed in wild-type HepG2 cells after 24-h treatment with the MTTP inhibitor CP-346086. These results suggest that defective release of lipoproteins from the ER membrane in VMP1-depleted HepG2 cells is not caused by MTTP inhibition. These results have been included in new Figure 6—figure supplement 1C and Figure 6—figure supplement 2C in the revised manuscript.

**Author response image 1. respfig1:** Interaction of VMP1 with APOB not MTTP is detected using the detergent 1% Triton-X100, but not dodecyl maltoside (1%)/cholesteryl hemisuccinate (0.2%). HepG2 cells were transfected with FLAG-VMP1 and lysed using lysis buffer containing 1% Triton-X100 (20 mM Tris-HCI, pH 8.0, 150 mM NaCl, 10% glycerol, 1% Triton-X100) or lysis buffer containing dodecyl maltoside (1%) and cholesteryl hemisuccinate (0.2%) (20 mM Tris-HCI, pH 8.0, 150 mM NaCl, 10% glycerol, 1% dodecyl maltoside [DDM], and 0.2% cholesteryl hemisuccinate [CHS]). Immunoprecipitation was performed using anti-FLAG M2 affinity gel. SDS-PAGE and immunoblotting was performed using indicated antibodies.

[Editors' note: the authors’ plan for revisions was approved and the authors made a formal revised submission.]

Reviewer 1's concerns are the most important:"I am still not convinced that VMP1 knockdown reduces the rate or extent of VLDL secretion from HepG2 cells. The evidence is inconsistent. Figure 4C, D suggests that there is a profound defect but is hard to reconcile with the data in Figure 4A, B and E, which suggest little change. In Figure 4E, the amount of APO proteins secreted after VMP knockdown is normalized to controls but it seems more appropriate to normalize it to the amount of APO remaining in the cells. When expressed in this way, there does not seem to be any decrease in the relative amount or rate of secretion of any of the APO proteins. How can this be reconciled with the evidence in Figure 4C, D?"

According to the reviewer’s suggestion, we re-evaluated lipoprotein secretion by calculating the ratio of the extracellular amount to intracellular amount of each protein (normalized to 0 h), and found that secretion of APOB and APOE, but not Albumin, was significantly reduced in VMP1-silenced cells (new Figure 4C). The secretion of APOA-I was only slightly reduced. These results suggest that a defect in secretion in VMP1-deficient cells is not general, but rather specific to lipoproteins (particularly APOB and APOE).

As we planned, we also performed a similar experiment in the presence of proteasome inhibitors to inhibit secondary degradation of intracellular apolipoproteins, but the results were not convincing probably because of cell death due to prolonged proteasome inhibition and, therefore, not included in the manuscript.

The inconsistency between the results in Figure 4A, B, E and 4C, D could be due to a difference in detection sensitivity of the methods used. Considering the fact that the defect in lipoprotein secretion was mild in epithelial cell-specific Vmp1 KO mice (Figure 3G), we think that the results in Figures 4A, B, E should correctly reflect the actual situation. Nevertheless, the HPLC data in Figures 4C, D still support our conclusion that there is a difference in the amount of secreted lipoproteins between control and VMP1-silenced cells. Thus, we have moved the HPLC data in original Figures 4C, D to new Figure 4—figure supplement 1A, B, and stated that different detection methods were used in new Figures 4A, B and new Figure 4—figure supplement 1A, B in the manuscript as follows; “Chromatographic analysis using different detection methods for neutral lipids also revealed significant reductions in lipoproteins”.

Reviewer 3 felt that the conclusions went beyond the actual data and need to be toned down and clarified (see below); they wrote, "The data indicate that VMP1 is not "essential" for APOB-lipoprotein assembly. If it was essential then we would have seen a more robust phenotype. It may play a role in lipoprotein secretion. But it is not a specific role. It probably plays a general role in secretion. This is based on the observation that there is reduction in APOB, APOA-I and APOE. This is not a new finding. The Authors should be realistic in interpretation of their data."

We agree that the phenotypes of Vmp1-deficient animals are milder than those of APOB- or MTTP-deficient animals. Thus, we have changed the title to "A critical role of VMP1 in lipoprotein secretion" to better reflect our results that VMP1 is not absolutely essential. Also, we have replaced the word "required" with "important" throughout the manuscript.

We did not claim that the function of VMP1 is specific to APOB assembly/secretion. As aforementioned, we have re-evaluated secretion of lipoproteins and confirmed that a defect in secretion in VMP1-deficient cells is not general, but rather specific to lipoproteins (new Figure 4C). In addition, we found that APOE was also trapped in large lipid structures in VMP1-silenced HepG2 cells (new Figure 6E). In contrast, APOA-I was not trapped in these lipid structures (new Figure 6F), which is consistent with the results of the secretion assay that the secretion of APOA-I was not much affected. Thus, we suggest in the text that defective secretion of APOB and APOE (new Figure 4C) is at least partly caused by trapping in the lipid structures (subsection “VMP1 is important for the release of lipoproteins from the ER membrane”, second paragraph).

We agree that VMP1 has a general role in secretion, as we mentioned in the previous version. It was shown in *Drosophila* S2 cells (Bard et al., 2006) and *Dictyostelium* (Calvo-Garrido et al., 2008). However, it is not shown in *Caenorhabditis elegans* or mammalian cells (Zhao et al., 2017). In *vmp1*-deficient zebrafish, formation of cartilage, which requires secretion of collagens, normally occurs and embryonic development was normal. Electron microscopy revealed the presence of lipid particles that are partially, but not completely, surrounded by ER membranes in VMP1-depleted cells (Figure 7), suggesting that neutral lipids were not completely released into the ER lumen. Furthermore, cells lacking factors required for budding from the ER to Golgi (e.g., TANGO1, TALI (Santos et al., 2016), cTAGE5 (Wang et al., 2016), or Surf4 (Saegusa et al., 2018)) did not show accumulation of similar large lipid-containing structures. Thus, these data suggest that, while VMP1 is important for general secretion in some cell types, the role of VMP1 in secretion is rather specific to lipoproteins in mammals and fish. However, as it is technically difficult to definitely demonstrate where each apolipoprotein and neutral lipids accumulate in the ER, we add the sentence “we do not exclude the possibility that VMP1 is also important at the step of ER-to-Golgi budding, which is not mutually exclusive”.

Reviewer #1:The authors have tried to respond with care to each of the reviewer comments. In looking at the rate of release of APOB and APOE, they find APOB levels decrease but a clear decrease for APOE. The legend does not explain how the gels were loaded and needs to be clarified. Also, it would be clearer for the reader if the authors graphed percent of total apolipoprotein in medium on a graph with time as a linear function.

We have now shown the results on a graph with time as a linear function in Figure 4C and included the information about the methods in the legend and Materials and methods as follows; “The medium was concentrated by TCA precipitation. Samples (approximately 7% or 14% volume of total precipitated media or cell lysates, respectively) were subjected to immunoblot analysis”.

Reviewer #3:– To prove definitely that VMP1 is essential for release of lipoproteins from the ER membranes, authors must follow APOB along with lipids. They should demonstrate that in the absence of VMP1, APOB remains in the ER membrane and is not present in the ER lumen. Thus, distribution of APOB and other control proteins in ER lumen and ER membrane must be studied. [Is there a way for you to account for APOB in support of your conclusions?]

We agree with this comment. However, in VMP1-deficient cells, the space between the ER membrane and lipids is very narrow (please see Figure 7D). In this case, it is very difficult to definitely prove that APOB indeed remains within the ER membrane bilayers but not present in the ER lumen by currently available methods (e.g., biochemical and immunoelectron microscopy). In fact, Dr. Fujimoto’s group previously showed the presence of APOB on similar structures (Ohsaki et al., 2008), but the data cannot tell whether APOB remains inside the membrane or in the lumen. We believe that our data showing that some of the lipid-containing structures are also positive for ADRP (please see Figure 6C, D) also suggest that these are not released into the lumen. We have mentioned this limitation in Discussion as follows; “However, as it is technically difficult to definitely demonstrate where each apolipoprotein and neutral lipids accumulate in the ER, we do not exclude the possibility that VMP1 is also important at the step of ER-to-Golgi budding, which is not mutually exclusive.”

– In absence of APOB-lipoprotein assembly, secretion of APOA-I is largely unaffected. Thus, significant decrease in APOA-I secretion in these studies suggests that VMP1 may have a generalized effect on secretion of soluble proteins as cited in the third paragraph of the Introduction. Thus, the argument that VMP1 is specific for APOB-lipoproteins is weak. APOA-I is secreted independent of APOB-lipoproteins. Its secretion does not require removal from ER membrane.

As mentioned above, we did not claim that the function of VMP1 in lipoprotein secretion is specific to APOB. In the re-evaluation of the secretion of lipoproteins, we found that secretion of APOE was also affected in VMP1-silenced HepG2 cells (new Figure 4C). The secretion of APOA-I was also affected but only slightly. As mentioned above, we have discussed the possible role of VMP1 in the ER-to-Golgi pathway (Discussion, first paragraph).

– Figure 5A: Is Sec61 surrounding the lipid droplet? It looks like LDs are in the lumen of ER. This suggests that they have been released from the ER membrane.

We agree that some but not all of large lipid-containing structures appear to be almost completely surrounded by Sec61B. However, the Sec61B signal is not always uniform; a significant number of these structures have Sec61B-weak regions (arrows in Author response image 2). Furthermore, APOB-weak or -negative regions are sometimes positive for ADRP (Figure 6C, D), suggesting that these are not released into the lumen. We have replaced the magnified images in Figure 5A with more representative ones and indicated Sec61B-weak regions.

**Author response image 2. respfig2:** Immunohistochemistry of the liver from 6-dpf *vmp1^-/-^* zebrafish using anti-Sec61B antibody and LipidTOX Red. Arrows indicate the regions where the Sec61B signals were weak. Magnified pictures of Figure 5A.

– EM is required to show that VMP1 deficiency leads to the accumulation of LDs and APOB (immune-gold) in the ER membrane if the hypothesis is that VMP1 is critical for the release of lipoprotein from the ER membrane. [Without such EM, can the conclusions be softened?]– Need separation of ER lumen and membrane and distribution of lipids and APOB in these fractions [if you want to conclude that the protein actually releases lipoprotein from ER membrane].

As aforementioned, it is difficult to prove that APOB is present on the ER membrane but not in the ER lumen by immunoelectron microscopy. It is also difficult to biochemically separate the ER membrane and ER lumen with intact localization of APOB. As mentioned above, we have discussed these limitations in the text (Discussion, first paragraph).

– It is likely that primordial lipoproteins are formed and they are in the ER lumen. VMP1 deficiency may interfere with their further transport to Golgi.

Although we cannot rule out this possibility, we think that it is less likely because most neutral lipids were not completely surrounded by the ER membrane (Figures 5A, B, and Figure 7).

– ADRP^+^ and APOB^+^ lipid droplets may be cytosolic droplets that contain APOB and may be unique to VMP1 deficiency.

Although we cannot rule out this possibility, we think that it is less likely. As the total number of ADRP-positive structures is significantly reduced (Figure 6C), the formation of cytosolic lipid droplets should also be defective in VMP1-depleted cells.

– This paper suggest that VMP1 is essential for the release of lipoproteins from the ER membrane, but the intestine-specific ablation phenotype on plasma is under whelming. Therefore, it is possible that VMP1 may play a role in lipoprotein secretion but it is not essential. Further, VMP1 deficiency appears to affect all apolipoprotein secretion. The authors are trying to over interpret their results. They should compare the phenotype in mice that are deficient in APOB and MTP; the two proteins known to be essential for B-lipoprotein assembly and secretion.

We agree that the phenotype of intestine-specific Vmp1 knockout mice is milder than that of intestine-specific Mttp knockout mice (Iqbal et al., 2013; Xie et al., 2006). We have added discussion regarding the difference in phenotype between these models as follows; “In contrast to intestinal epithelial cell-specific Mttp-deficient mice (Iqbal et al., 2013; Xie et al., 2006), intestinal epithelial cell-specific Vmp1-deficient mice showed milder phenotypes in lipoprotein secretion; the level of serum triglyceride did not decrease in intestinal epithelial cell-specific Vmp1-deficient mice (Figure 3G). Thus, although VMP1 is important, it is not absolutely essential for lipoprotein secretion.” Also, we have changed the manuscript title to better reflect our results and replaced the word "required" with "important" throughout the manuscript.

– It is unclear why LDL and HDL are decreased. How did they measure LDL and HDL? Why cholesterol decreased but no change in triglyceride? This phenotype does not support the idea that VMP1 is essential for APOB-lipoprotein assembly and secretion. Most likely VMP1 is involved in general secretory pathway. Its deficiency slows down secretion of APOB. Similar thing is happening for APOA-I and APO**E**. Thus, there is little specificity to this process.

We have measured LDL and HDL in mice using an OLYMPUS AU480 automatic biochemical analyzer equipped with standard reagents that can selectively separate LDL or HDL from other lipoproteins and quantify the amount of LDL or HDL cholesterol, according to the manufacturer's explanation. We don’t know the reason why levels of triglyceride were preserved in intestinal epithelial cell-specific *Vmp1*-deficient mice, but it is likely due to a possible compensatory mechanism, which will be investigated elsewhere in the future. As we discussed above, we think that VMP1 is not absolutely essential for lipoprotein secretion and is not specific to APOB.

– VMP1 also affects SERCA. Can the phenotype be explained by alterations in calcium pump in the ER?

We also thought this possibility, but ADRP^+^ and APOB^+^ crescent structures were not formed in cells treated with thapsigargin, a SERCA inhibitor (Figure 6—figure supplement 2B). Thus, the phenotypes by VMP1-deficiency cannot simply be explained by SERCA inhibition.

– Was blood collected from fasted mice?

As we have mentioned in the figure legend and Materials and methods in the original version, we used mice fed ad libitum.

– Figure 3: Why there is a significant decrease in HDL? If VMP1 plays a role in APOB-lipoprotein assembly then it should not affect HDL levels. These data demand fat absorption studies in these mice to see if intestinal deficiency of VMP1 really has any significant defect in lipid absorption.

We actually do not have any idea why HDL was reduced in *Vmp1*-deficient mice. Further analysis will be required to address this question using intestinal epithelial cell-specific *Vmp1*-deficient mice, which will take more than one year for us. We think that such detailed lipid analysis is beyond the scope of this first cell biological report.